

# A component based modular treatment of the soil-plant-atmosphere continuum: the GEOSPACE framework (v.1.2.9)

Concetta D'Amato[1,2], Niccolò Tubini[2], and Riccardo Rigon[1,2]

[1]Center Agriculture Food Environment - C3A, University of Trento, Trento, Italy
[2]Department of Civil, Environmental and Mechanical Engineering - DICAM, University of Trento, Trento, Italy

**Correspondence:** Concetta D'Amato (concettadamato94@gmail.com)

**Abstract.** The soil-plant-atmosphere continuum (SPAC) system is a complex and interconnected network of physical phenomena, encompassing heat transfer, evapotranspiration, precipitation, water absorption, soil water flow, substance transport, and gas exchange. These processes govern the exchange of energy, matter, and water within the SPAC system. To better understand and model SPAC interactions, interdisciplinary approaches are essential due to the inherent complexity of the system. Instead of relying on a single monolithic model, we propose a component-based modeling approach, where each component addresses a specific aspect of the system. Object-oriented programming (OOP) is adopted as the foundational framework for this approach, providing flexibility and adaptability to accommodate the ever-changing nature of the SPAC system.

The Soil Plant Atmosphere Continuum Estimator in GEOframe (GEOSPACE) is presented in this paper, in particular the one-dimensional development, GEOSPACE-1D. The framework is a tool designed to facilitate robust, reliable and transparent simulations of SPAC interactions. It embraces the principles of open-source software and modular design, aiming to promote open, reusable, and reproducible research practices. By implementing the OOP, GEOSPACE-1D breaks down the complexity of SPAC modeling into smaller, self-contained structures, each responsible for a specific scientific or mathematical concept. This modular architecture adheres to the "open to extensions, closed to modifications" philosophy, enabling easy model extension without disrupting existing components. Equations are implemented in an abstract manner, emphasizing the use of common interfaces over concrete classes, a hallmark of contemporary OOP. GEOSPACE-1D adopts a generic programming framework, where distinct classes adhere to a common interface. This compartmentalization serves two critical purposes: validating individual processes against analytical solutions and facilitating the integration of novel processes into the system.

The paper emphasizes the significance of modeling the coupling between infiltration and evapotranspiration for accurate hydrological simulations. It explores the interplay between plant transpiration, soil evaporation, and soil moisture dynamics, highlighting the need to account for these interactions in SPAC models. The paper concludes by underlining the importance of modularity, transparency, and openness in SPAC modeling, principles that underlie the development of GEOSPACE-1D and its components. Overall, GEOSPACE-1D represents a promising approach to SPAC modeling, providing a flexible and extensible framework for studying complex interactions within the Earth's Critical Zone. It is worth recalling that the fundamental premise of GEOSPACE-1D is not to create a single soil-plant-atmosphere model, but to establish a system that allows the creation of a series of soil-plant-atmosphere models, adapted to the specific needs of the user's case study.



# 1   Introduction

The Soil-Plant-Atmosphere Continuum (SPAC) encompasses a wide range of interconnected physical phenomena, including heat transfer, evapotranspiration, precipitation, water absorption and infiltration, soil water flow, substance transport, and gas exchange, all of which influence the exchange of energy, matter, and water among these three compartments (Fisher and Koven, 2020; Blyth et al., 2021; Li et al., 2021). A variety of formulations for the underlying physics of these processes are currently debated, including soil-root interactions (Steudle, 2000; Schröder et al., 2008; Manoli et al., 2017), alternative formulations for plant hydraulics (Verhoef and Egea, 2014b; Silva et al., 2022; Giraud et al., 2023), constraints imposed by water-limited availability (water stress) and their combinations (Lhomme, 2001; Verhoef and Egea, 2014b), the characterization of soil properties in the presence of roots (York et al., 2016; Carminati and Javaux, 2020), and coupling plant behavior with atmospheric transport (Katul et al., 2001; Poggi et al., 2004; Mauder et al., 2020; Finnigan et al., 2009). Additional discussions include the statistical description of plant canopies (Kerches Braghiere, 2018; McGrath et al., 2016), individual plant traits (Mencuccini et al., 2019; Cranko Page et al., 2024), and the interactions between trees, soil microbiology (Cassiani et al., 2015; Simard et al., 1997), and atmospheric processes (Brunet, 2020). Depending on specific objectives and the temporal and spatial scales of analysis, various simplifications of these processes are often employed (Anderson et al., 2003; Donovan and Sperry, 2000).

Given the complexity of the SPAC domain, numerous modeling approaches have been developed. These include physically based models (PBM) (Fatichi et al., 2016) and those leveraging statistical learning techniques, such as machine learning (ML) (Pal and Sharma, 2021). Traditional PBM-based land surface models, widely used in hydrology and agronomy, often employ simplified governing equations, such as the Penman-Monteith equation (Pereira et al., 2015) or the Priestley-Taylor approach (Formetta et al., 2014). More advanced SPAC models include SVAT (Soil-Vegetation-Atmosphere Transfer) models and LSM (Land Surface Models). While SVAT models focus on vegetation-related processes and LSMs cover a broader range of processes, the distinction between these categories is not always clear-cut. Comprehensive reviews of such models are available in Blyth et al. (2021), Fisher and Koven (2020), and Pal and Sharma (2021).

Achieving a deeper understanding and accurate modeling of SPAC interactions requires highly interdisciplinary approaches. This necessitates moving beyond the traditional notion of a "model."

To address the implementation challenges arising from the complexity of SPAC processes and the diversity of possible solutions, the literature advocates dividing software into self-contained, independent components interconnected through a supporting software layer. This approach, known as "Modeling by Components" (MBC), has been in use for over forty years (Holling, 1978) and was initially developed to integrate knowledge across disciplines (Moore and Hughes, 2017). In recent decades, MBC has gained significant traction within environmental modeling (Argent, 2004; Serafin, 2019). Notable MBC implementations in hydrology and meteorology include TIME (Rahman et al., 2003), CSDMS (Peckham et al., 2013), ESMF



(Collins et al., 2005), OMS (David et al., 2013), and RAVEN (Craig et al., 2020). A more extensive list can be found in Chen et al. (2020).

True MBC systems adopt a service-oriented architecture (SOA) (Richards and Ford, 2020), which facilitates the integration of heterogeneous data sources. SOA frameworks are inherently scalable, designed to operate across diverse machines and architectures, and are foundational to Digital Earth infrastructures (Rigon et al., 2022). By abstracting computational details, SOA enables users to focus on modeling rather than the complexities of the underlying computational engines.

Despite the advantages of MBC concepts, practical implementation remains challenging. It often requires programmers to
adopt new workflows and habits. The OMS framework has explicitly addressed these challenges, bridging the gap between conceptual elegance and practical usability (Lloyd et al., 2011). These frameworks aim to meet broader scientific needs while adhering to good scientific practices, as outlined in Rigon et al. (2022).

To our knowledge, no existing SVAT or LSM models implement a true MBC structure, although some exhibit highly modular software organization. Beyond leveraging MBC, incorporating internal modularity through Object-Oriented Programming
(OOP) principles is equally important, as emphasized by Rouson et al. (2014) and Gardner and Manduchi (2007). OOP promotes code reuse, improves readability, and enhances efficiency and scalability. Moreover, employing appropriate levels of abstraction, as theorized by Berti (2000a), enhances model adaptability and reliability while minimizing the need for modifications to existing code.

These principles have guided the development of GEOSPACE, the Soil-Plant-Atmosphere Continuum Estimator in GE-
Oframe. This software platform integrates MBC concepts with OOP principles to transcend traditional modeling paradigms, focusing on the interactions and feedback mechanisms within the SPAC.

This paper is organized as follows: Section 2 provides an overview of the GEOSPACE system and its hierarchical software architecture. Sections 3, 4, and 6 introduce its main components: WHETGEO, GEOET, and BrokerGEO. Section 5 discusses the implementation of the stress function, addressing evapotranspiration constraints caused by water scarcity or other envi-
ronmental factors. Each section includes a concise overview of the mathematical equations employed and relevant software implementation details. Readers less interested in the informatics can skip the latter portions of these sections. Section 7 presents a use case, followed by Section 8, which describes the availability of the code, executables, training materials, fair use conditions, and concludes the paper.

While this paper does not delve into the rationale behind the implemented physics, which is addressed in other contributions,
it focuses on software organization and the integration of components into a cohesive modeling solution. This approach, by introducing feedback mechanisms, transcends traditional modeling paradigms to become more than a single model with boundary conditions (Staudinger et al., 2019). Appendices provide technical details, and supplementary materials include notebooks for data preparation, output visualization, and video tutorials.





## 2   GEOSPACE-1D System Overview and its perceptual model

The framework presented here, the Soil-Plant-Atmosphere Continuum Estimator in GEOframe (GEOSPACE), is an ecohydro-logical framework within the GEOframe system. It is designed to simulate interactions within the soil-plant-atmosphere continuum and analyze processes occurring in the Earth's Critical Zone (CZ) (National Research Council et al., 2001). GEOSPACE-1D models the mass and energy budgets as well as water flow along a soil column, accounting for water uptake by vegetation as evapotranspiration flux (ET).

As outlined in the Introduction, the framework offers multiple alternatives for simulating key physical processes (e.g., evapotranspiration, stress factor computations, soil parameterizations) with varying levels of complexity and detail. This flexibility allows users to tailor the model to their specific case studies, compare different formulations, and easily add new features. Such modularity enhances the reliability of modeling solutions by enabling the integration of appropriate components. The component-based structure of GEOSPACE-1D also improves software robustness and facilitates third-party testing and inspec-

tion. Moreover, the software is open source and adheres to modern software engineering practices (Rigon et al., 2022). The OMS3 workflow is recorded in ".sim" files, ensuring that any deterministic simulation can be precisely replicated. GEOSPACE-1D prioritizes the development of reliable, robust, and replicable models, as emphasized in Prentice et al. (2015). Its flexibility allows for varying degrees of realism by selecting or developing components aligned with modeling objectives and advancements in research.

Based on the principles of Tubini et al. (2021), the implementation of the basic equations of GEOSPACE-1D is abstract. The equations describing the processes inherit from a common interface, following the principle of "programming to interfaces, not to concrete classes." These equations are organized into libraries that streamline the implementation of partial differential equations (PDEs), ordinary differential equations (ODEs), and other equation types with flexibility and minimal effort.

While the ultimate goal is to comprehensively cover all compartments of the SPAC, this paper primarily focuses on water

and energy exchanges between soil, plants, and atmosphere, with a reasonable treatment of canopies. The GEOframe system already includes a wide array of ODE-based models for some of these exchanges. However, GEOSPACE-1D specifically incorporates PDEs, particularly for the soil compartment. It features a robust implementation of the Richards-Richardson equation (Tubini et al., 2021) for 1D soil water flow and extends the Penman-Monteith approach for transpiration (Schymanski and Or, 2017; Bottazzi et al., 2021). Additionally, ancillary components have been developed for radiative transfer based on

Ryu et al. (2011) and de Pury (1995), and for coupling the water budget with the energy budget and solute transport in the soil.

GEOSPACE-1D comprises a coupled model consisting of three primary components: WHETGEO, GEOET, and BrokerGEO. WHETGEO, Water Heat and Transport in GEOframe (Tubini and Rigon, 2022), solves the conservative form of Richardson-Richards equation using the Newton-Casulli-Zanolli algorithm (Casulli and Zanolli, 2010), and also implements a numerical solution to solve the transport equation adopting the algorithm presented in Casulli and Zanolli (2005).

GEOET (GEOframe EvapoTranspiration) is a suite of models that is designed to implement different formulations of ET, from the simplest Priestley-Taylor (PT) formula (Priestley and Taylor, 1972) to the complex computation of the energy budget at the canopy scale, as described in Rigon and D'Amato (2024).





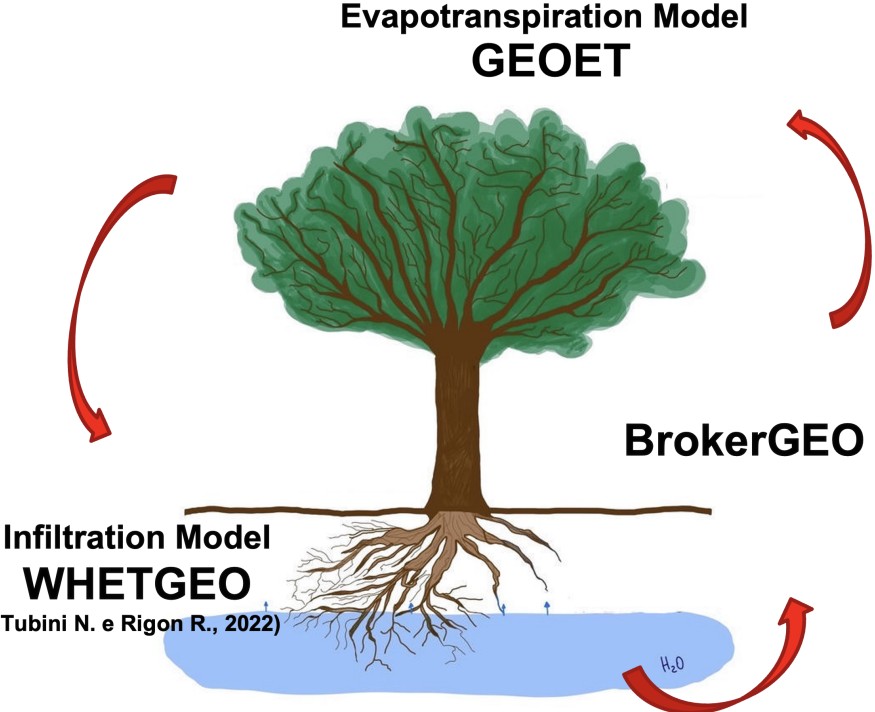

**Figure 1.** GEOSPACE-1D cycle path: Presented graphically (designed by D'Amato C.) is the cyclic operation of GEOSPACE-1D, highlighting the crucial linkage among its three components. This feedback mechanism plays a key role in ensuring the mass and energy balance throughout the system.

Currently GEOET, in addition to PT, incorporates the Penman-Monteith FAO model (PM-FAO) and the GEOframe-Prospero model (Bottazzi, 2020; Bottazzi et al., 2021). It is also designed to integrate more complex models that account for plant
hydraulics, being implemented according to Rigon and D'Amato (2024). The presence of multiple models for computing evapotranspiration within a unified framework enables a comprehensive comparison of models and parameterizations, as they can leverage common auxiliary components. This possibility also meets the needs of users by offering a selection of modelling approaches of different level of complexity, allowing users to choose according to their specific needs and available data. GEOET is also designed to implement the multiple water and environmental stress functions mentioned in the introduction,
which are currently based on the Jarvis model (Macfarlane et al., 2004) and the Medlyn stomatal conductance model (Medlyn et al., 2011; Ball et al., 1987; Lin et al., 2015). GEOET also models depth growth and root density functionally to understand soil-plant interactions in the process of root water uptake.

Finally, BrokerGEO is the coupler that allows the exchange of data between the other two components in memory, splits evaporation ($E_s$) and transpiration ($E_l$) between the control volumes in which the soil column is discretized.





The operational flow of the model follows a cyclic pathway, as illustrated in Figure 1. Starting with WHETGEO-1D, GEOSPACE-1D computes the water suction for each control volume within the soil column. This information, integrated into the GEOET-StressFactor modules (further described in Section 5), plays a crucial role in determining the reduction in ET for each control volume, which is then consolidated into a global water stress factor value.

    Furthermore, the GEOET-StressFactor modules incorporate additional reductions in ET associated with environmental variables, including air temperature, net radiation, and vapor pressure deficit, when necessary, in accordance with the selected

method. The specific manner in which these stress factors are utilized varies depending on the chosen method or ET model, as detailed in Section 4, with the objective of limiting water abstraction from the soil.

    Subsequently, BrokerGEO is responsible for partitioning the global AET into the soil control volumes using a root functioning model. Finally, an iterative process begins, during which WHETGEO, informed by the water evapotranspired from each

control volume, recalculates the soil water potential, ensuring the conservation of both mass and energy budgets.

### 2.1   General notes about the software organization of GEOSPACE-1D

The macro conceptual structure of GEOSPACE-1D is mirrored in a hierarchy of software entities. At a higher level are the OMS3 components (David et al., 2013, 2014). OMS3 is a component-based environmental modeling framework that empowers developers to create distinct components for individual modeling concepts. These components can be, in principle executed

and tested independently, thus establishing what could be called a secondary level of concern. A graphical example of a component is illustrated in Figure 2. Each input parameter is sourced from a reader component or another component if derived from some modelling, while each output parameter is handled by a writer component. This setup facilitates straightforward management of data formats, streamlining the process for ease of use. Leveraging the modularity of OMS3, GEOSPACE-1D components integrate at runtime by connecting them with the OMS3 DSL language based on Groovy (https://groovy-lang.org,

last accessed: December 19, 2024). A standard operational configuration of the GEOSPACE-1D OMS3 components during runtime is depicted in Figure 3, with simplification achieved by omitting all input/output connections for clarity.

    The GEOSPACE-1D system depicted in Figure 3 consists of three main interconnected parts, each comprising a set of related components. WHETGEO actually has multiple variants, each distinguished by unique capabilities outlined in Tubini and Rigon (2022). More intricate is the GEOET, comprising five distinct components, each tasked with specific functions such

as stress factor estimation, roots modeling, soil evaporation estimation, and transpiration computation using, in this case, the Prospero model. Alternatively, a single component can replace the latter two and the one estimating the total evapotranspiration by employing, for instance, the Priestley-Taylor formula to estimate comprehensive evapotranspiration values. BrokerGEO employs two components in its connector role. The arrows denote variable flow among components, with thickness reflecting the volume of exchanged variables.

OMS3, beyond the components, offers essential services, including tools for calibration and implicit parallelization of component runs. Further insights into the framework are available in the Supplemental material.

    Deeper within the software, written in Java programming language, it is organized into `packages` that encapsulate cohesive functional modules. The names of the packages formed are as follows:



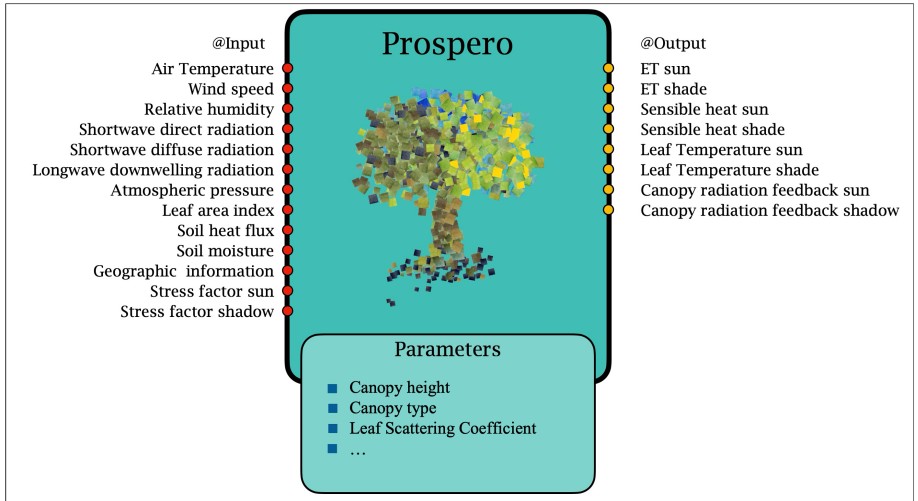

**Figure 2.** A pictorial representation of the Prospero OMS3 component with most of its inputs and outputs.

❏ `it.geoframe.blogspot.`**`modelname.packagename`**

and are designed from being confused with that of other models in the literature that implement the same classes. The site `https://geoframe.blogspot.com/` is the blog where all the news and material about GEOframe are posted and the uniqueness of the URL guarantees that the name of the packages and classes are unique over the web. The contents of these packages are illustrated, for each of the three compartments, in the following sections.

A few software engineering choices were made and are characteristics "patterns" of the programming in GEOframe. Classes represent the finer programming level. In fact, internally, GEOSPACE-1D is coded with the Java language by organizing the various classes between interfaces and abstract classes. This adheres to the object-oriented principle "program to interfaces, not concrete classes," facilitating the creation of scalable and maintainable software solutions. As stated earlier, re-usability of the Java code is one of the prerogatives of the model. Therefore, we have adopted a generic programming approach (Berti, 2000b) of decoupling the concrete data representation from the algorithm implementations, by balancing it with specific approaches to improve the computational efficiency of the software. Because of these special programming structures, "patterns", are often used. In particular the so called `Factory Pattern` (Gamma et al., 1995a) is used to instantiate at run time the concrete classes chosen by the user among the various possibilities.

Because we are mostly interested in the implementation issues in this paper, it merits to acknowledge a crucial aspect for facilitating information exchange between the soil and atmospheric compartments which is how the data classes, i.e. the Java classes that manage the data quantities, are defined. These include:

– `ProblemQuantities`

– `InputTimeSeries`







**Figure 3.** Here is an illustration of a simplified configuration in GEOSPACE-1D, focusing on component organization. We have omitted the components responsible for file I/O. One of the components, WHETGEO, exemplifies the various possibilities, differing in whether they account for the influence of the energy budget on flow.





    – `Parameters`

These data classes serve as data containers and are managed distinctively from conventional OOP practices, but similar to
traditional scientific programming. They are mutable singleton static classes, instantiated once and serving as repositories for
data updated at each time step. Further elaboration on these classes will be provided in the following sections.

## 3 WHETGEO

The first pillar of the GEOSPACE-1D modeling system is WHETGEO (Tubini and Rigon, 2022; Tubini et al., 2021). Its 1D
deployment, WHETGEO-1D is a new physically based model, simulating the water and energy budgets in a soil column.
It solves the conservative form of Richardson-Richards equation, $R^2$, (Richards, 1931; Richardson, 1922) using the Newton-
Casulli-Zanolli (NCZ) algorithm (Casulli and Zanolli, 2010) and also implements the numerical solution to solve the advection-
dispersion equation adopting the algorithm presented in Casulli and Zanolli (2005), currently applied to the heat transport.
More comprehensive information about WHETGEO can be found in Tubini (2021), Tubini and Rigon (2022) and Tubini et al.
(2021).

WHETGEO delves into infiltration and soil moisture dynamics that have cascading effects on various aspects of the envi-
ronment, water resources, accurate water balance estimation, aquifer recharge prediction and flow patterns.

For what regards evaporation, it is well-established that plant productivity is significantly influenced by the patterns of soil
moisture dynamics (Porporato et al., 2004). Soil moisture deficit, in particular, reduces plant water potential, inducing wa-
ter stress, which can lead to dehydration, loss of turgor, xylem cavitation, stomatal closure, and a decrease in photosynthesis
(Nilsen and Orcutt, 1996). At the same time, it is an oversimplification to model soil moisture dynamics without considering
transpiration, which constitutes a significant portion of the water budget. The relationship between soil moisture and evapo-
transpiration is a crucial component for accurately representing soil water balance. Regardless of the specific ET model used,
it is the amount of water extracted from the soil at various depths by plant roots (Evaristo and McDonnell, 2017). These roots
are capable of absorbing substantial amounts of water, significantly altering the distribution of the water column.

An approach to include the evapotranspiration flux within the soil moisture dynamics is to add a sink term representing water
extraction by plant roots in the $R^2$ equation, obtaining a modified one-dimensional continuity equation (Feddes et al., 1976;
Molz, 1981):

$$\frac{\partial \theta}{\partial t} + S_s \frac{\theta}{\theta_s} \frac{\partial \psi}{\partial t} = \nabla \cdot (K(\theta)\nabla(\psi + z)) - S(z) + R(z) \tag{1}$$

where the forces acting are gravity $z$ [L], and the matric potential $\psi$ [L]. In Eq. 1, $K$ [LT$^{-1}$] is the hydraulic conductivity; $\theta$
[−] is the dimensioneless volumetric water content; $\nabla$ [L$^{-1}$] is the gradient operator; $z$ [L] is the vertical coordinate, positive
upward, $S$ is the water extraction function [T$^{-1}$], and $R$ [T$^{-1}$]is the water redistribution function by roots. The function
$S(z)$ represents water extraction by plant roots and can depend on space, time, root-density distribution, water potential, water
content, or a combination of these variables (Feddes et al., 1976; Perrochet, 1987; Lai and Katul, 2000). $S_s$ [L$^{-1}$] is the specific





storage coefficient, defined as

$$S_s := \rho g(n\beta + \alpha) \tag{2}$$

with $\rho$ [ML$^{-3}$] being the water density, $g$ [LT$^{-2}$] is the gravitational acceleration, n [L$^3$L$^{-3}$] is the soil porosity, $\beta$ [LT$^2$M$^{-1}$] is the liquid compressibility, and $\alpha$ [LT$^2$M$^{-1}$] is the soil matrix compressibility.

Molz and Remson (1970) highlight the impracticality of modeling water transport in soil with complex root systems considering flow to individual rootlets. Precise root geometry is hard to measure and varies over time. Additionally, root water permeability changes along their length, noted by Kramer (1970). Consequently, the extraction functions $S(z)$ is treated with simplified models which adopt a macroscopic, not microscopic approach, which is actually computed by the BrokerGEO components (described in section 6).

As the source term models, the roots themselves, are capable of redistributing water between different soil layers (Beyer et al., 2018) but the modelling of the source term $R(z)$ is not under scrutiny in this paper.

### 3.1 Extension of the Richards Solver, i.e. on the RichardsRootSolverMain

The algorithmic concepts of WHETGEO are comprehensively described in Tubini and Rigon (2022), but here, we summarize only those aspects, which are related to the extension we made for allowing the connection with GEOET and BrokerGEO.

In WHETGEO, the column of soil is discretized in layers, parts of the soil column with the same soil type, and control volumes, finite-volume elements, in which the R$^2$ equation is solved. Each control volume is characterized by geometrical quantities, a parameter set, containing all the parameters that control the dynamics of the flow, the form of the equations to be solved, which are specific for each control volume. All of this information is stored in the grid file and, once read, is stored in the `Geometry` and `ProblemQuantities`, singleton classes as illustrated in Figure 4.

Upon a more detailed examination of Figure 4 reveals that `ProblemQuantities` and `Geometry` are typical example of "reflexive association", a type of relationship between elements in a class diagram where an element is associated with itself. In other words, it is an association between instances of the same class and it means that the class could work stand-alone. The term $S(z)$ in Eq. 1 is a sink which affects any layer and this can be obtained by means of the introduction of some new classes.

The first class that has been added is:

– `ComputeQuantitiesRichardsRoot`

`ComputeQuantitiesRichardsRoot` is a java class with the responsibility of managing the evapotranspiration demand for each control volume, sourced from BrokerGEO. Its primary function involves assessing the requested water volume against the available water content in each control volume, thereby estimating the reduction in ET and deriving a stress factor, $g_{w_i}$, for every soil layer. These stress factors are subsequently utilized iteratively to determine the actual evapotranspiration (AET) through information exchange with GEOET via BrokerGEO. AET is then deducted from each control volume, with the class overseeing algorithm convergence as well.

A closer inspection of Figure 4 reveals that the `ComputeQuantitiesRichardsRoot` class is composed by aggregation with the `ProblemQuantities` class and the `Geometry` class as mentioned above. The first one contains all the variables







**Figure 4.** UML class diagram for the `RichardsRootSolver1DMain` class. The class solves Eq. 1 directly calling the methods of the concrete classes `ComputeQuantitiesRichardsRoot` and `ComputeQuantitiesRichards`.





of the models and its implementation using the singleton pattern (Freeman et al., 2008), whereas, the second one manages the geometric features of the grid and how grid elements are connected to each other.

While the present deployment of GEOSPACE works in a 1D column, however it is ready to manage the more complex topologies of a 2D or a 3D future versions of the system.

The relationship between the classes `ComputeQuantitiesRichardsRoot` and `ProblemQuantities` can be described as a combination of both "association" and "aggregation" (Fowler, 2004). The `ComputeQuantitiesRichardsRoot` class maintains a reciprocal "association" with the `ProblemQuantities` class through the instantiated variables allowing them to interact and exchange information. Furthermore, an "aggregation" relationship exists where the `ComputeQuantities-RichardsRoot` class encapsulates an instance of the `ProblemQuantities` class to manage and compute various problem-related quantities as is illustrated by an empty diamond shape (Figure 4). Overall, this relationship structure enhances the modularity and organization of the software design, enabling the `ComputeQuantitiesRichardsRoot` class to efficiently utilize and manage the data provided by the `ProblemQuantities` class.

The class involved in solving Equation 1 considering the amount of water removed by evapotranspiration is the concrete class, `RichardsRootSolver1DMain`, as shown in the UML diagram of Figure 4.

The `RichardsRootSolver1DMain` class, as shown in Figure 4 is directly connected with `ComputeQuantitiesRichards-Root` and `ComputeQuantitiesRichards` and with their methods to compute the solution of the pressure value $\psi$ at any time step. In this specific case, the `RichardsRootSolver1DMain` class uses the `ComputeQuantitiesRichardsRoot` and the actual solver is the `solve()` method which contains the solving algorithms. This abstract structure allows with the change of each of the "`Compute`" classes to change the solver type by just adding some new class in substitution.

## 4 GEOET

The process of plant transpiration propels the exchange of water and energy between the Earth's surface and the atmosphere (Katul et al., 2012). This phenomenon significantly impacts the uptake of carbon by ecosystem and also plays a pivotal role in determining how rainfall infiltrates into the soil and the moisture profile dynamic. In fact, the interaction between soil evaporation and plant transpiration is not merely a sum of physical processes, but is influenced by feedback mechanisms. As plants transpire, they create a suction that draws moisture from the soil into their root systems, thereby influencing the rate of soil evaporation. Conversely, soil evaporation can reduce the available moisture for plant roots, impacting their ability to transpire effectively. This dynamic coupling shapes the moisture profile within the soil, significantly changing the overall water availability for the soil and vegetation and for the entire hydrological cycle.







**Figure 5.** UML class diagram for the `Parameters` class showing the relation of aggregation and association with the ET models solvers and the input reader class.



**The Priestley-Taylor $E_T$ estimator**

The GEOET system incorporates four evapotranspiration models, as illustrated in Figure 6. Starting with the simplest model, i.e., the widely used Priestley-Taylor model (PT) (Priestley and Taylor, 1972), which is based on the formula:

$$ET_{PT} = \alpha \frac{(R_n - G)\Delta}{(\Delta + \gamma)} \tag{3}$$

where: $\alpha$ is an empirical coefficient relating actual evaporation to equilibrium evaporation, $\Delta$ is the slope of the saturation vapor pressure and air temperature curve [kPa C$^{-1}$], $\gamma$ is the psychrometric constant [kPa °C$^{-1}$], $R_n$ is the net radiation [W m$^{-2}$] and G is the ground heat flux [W m$^{-2}$].

The implementation of this formula is relatively straightforward, although the radiation term requires a more detailed and careful evaluation. Further information about this equation can be found in Appendices A.

**The Penman-Monteith FAO estimator**

The second model at present implemented is the Penman-Monteith FAO approximation (PM) (Penman and Keen, 1948), an adaptation of the Penman-Monteith model, as outlined in the following equation:

$$ET_0 = \frac{1}{\lambda} \frac{0.408\Delta(R_n - G) + \gamma\frac{900}{T+273}u_2\delta_a}{\Delta + \gamma(1 + 0.34u_2)} \tag{4}$$

where, $ET_0$ is the reference evapotranspiration [mm day$^{-1}$], $R_n$ is the net radiation at the crop surface [MJ m$^{-2}$ day$^{-1}$], $G$ is the soil heat flux density [MJ m$^{-2}$ day$^{-1}$], $T$ is the mean daily air temperature at 2 m height [°C], $u_2$ is the wind speed at 2 m height [m s$^{-1}$], $e_s$ is the saturation vapour pressure also known as vapor pressure at the dew point [kPa], $e_a$ [kPa] is the actual vapour pressure , $\delta_a := (e_s - e_a)$ [kPa] is the saturation vapour pressure deficit, $\Delta$ [kPa °C$^{-1}$] is the derivative of the Clausius-Clapeyron formula.

In terms of computational complexity, these models are relatively straightforward. Both PT and PM are equipped with dedicated packages housing their respective core Java classes for solving the equations, `PriestleyTaylorModel.java` and `PMFAOModel.java`. Additionally, each package includes various 'Solver' classes designed to invoke methods from the main model class, enabling the computation of solutions that account for environmental inputs and stress factors.

Specifically, to ensure that users do not input more information than necessary or provide data unrelated to their chosen method, we differentiate between solvers for calculating potential evapotranspiration (without stresses) and actual evapotranspiration (with the possibility to select among the various type of stresses).

**The Prospero Model**

The Prospero model (Bottazzi, 2020; Bottazzi et al., 2021), is a physically based approach for calculating transpiration. The transpiration, $E_l$, is considered for the sunlit and shaded fractions of the canopy while the soil evaporation, $E_s$, is estimated using the residual radiation hitting the soil. $E_s$, at present, is determined according the FAO Penman–Monteith model, i.e. $E_s = ET_0$. Meanwhile, $E_l$ is computed using Prospero model that implements a modified version of the Schymanski and





Or (SO) model (Schymanski and Or, 2017), which has been upscaled to address canopy-level transpiration and ensure mass conservation during periods of water stress.

As described in Appendix A, SO solves the stationary energy budget coupled with the water vapor transport and sensible heat transport in a zeroth-order approximation, with the help of information derived from the Clausius-Clapeyron equation. Furthermore, it returns, not only a $E_l$ estimate but also the sensible heat $H$ and vapor pressure gap $e_\Delta$, besides the temperature of the leaves, $T_l$. The latter variable is the key for obtaining the others, as below:

$$T_l = \frac{R_n + a_{sH}A_{tr}\epsilon_l \sigma T_a^4 + c_H(a_{sH}, A_{tr}) \cdot T_a + c_E(a_{sE}, A_{tr}, g_s) \cdot (\Delta \cdot T_a + \delta_a)}{c_H(a_{sH}, A_{tr}) + c_E(a_{sE}, A_{tr}, g_s)\Delta + a_{sH}A_{tr}\epsilon_l \sigma T^3} \tag{5}$$

where: $R_n$ is the the net radiation, $a_{sH}$ is the sides of surface exchanging sensible heat and longwave radiation, equal to 1 for single-layer exchange, 2 for two-layer exchange such as leaves[-], $A_{tr}$ is the transpiring surface for unit of ground surface [-], $\epsilon_l$ [-] is the leaves emissivity, $\sigma = 1.67\,10^{-8}$ [$\mathrm{Wm^{-2}K^{-4}}$] is the Stefan-Boltzmann constant, $a_{sE}$ represents the sides of surface exchanging latent heat, equal to 1 for hypostomatous, and 2 for amphistomatous [-], $g_s$ is the stomatal conductance [$\mathrm{ms^{-1}}$], $\delta_a = P_{ws}$ and $P_w$ are the saturation water vapour pressure and the water vapour pressure, respectively, $c_E$ is the total conductance for water vapor evapotranspiration transport [$\mathrm{ms^{-1}}$] and $c_H$ is the total conductance for the sensible heat transport [$\mathrm{ms^{-1}}$].

It is assumed that the right-hand-side terms in Equation 5 are all known. Furthermore, the water pressure gap is estimated as follows:

$$e_\Delta := e_s - e_l = \Delta(T_l - T_a) + \delta_a \tag{6}$$

The transpiration is calculated as:

$$E_l = c_E(a_{sE}, A_{tr})e_\Delta; \tag{7}$$

Finally, the sensible heat is computed as:

$$H_l = c_H(a_{sH}, A_{tr})(T_l - T_a) \tag{8}$$

For further information on these formulas or solutions, please refer to Bottazzi (2020), Bottazzi et al. (2021) or Rigon and D'Amato (2024).

## 4.1 The GEOET informatics organization

The evapotranspiration component, GEOET, developed as part of this paper, is based on its precursor, GEOframe-ETP model (Bottazzi, 2020; Bottazzi et al., 2021) whose original source code is available at *https://github.com/geoframecomponents/ETP*. Both GEOframe-ETP and GEOET simulate the evapotranspiration according to different evapotranspiration models: the Priestley-Taylor model (Priestley and Taylor, 1972), the Penman-Monteith FAO model (Allen et al., 1998), and the GEOframe-Prospero model (Bottazzi, 2020; Bottazzi et al., 2021). But, in moving from one software to the other, the refactoring of the existing codes, was substantial both at design level and algorithmic level, as you can see in the box diagrams of Figure 6. The





reorganization and the re-engineering of the software were essential to allow the use of multiple options of evapotranspiration physics, to introduce a separate component for calculating the stress factor, making possible to apply them to all evapotranspiration models, and facilitating the connection of any of the ET components with other model components and particularly enabling its linkage with WHETGEO.

The new version of the code thus allows the physical processes of evapotranspiration to be analysed individually and also in
conjunction with the infiltration process. To obtain this, as shown in Figure 6, all the evapotranspiration models were separated into packages of classes with specific tasks. Furthermore, depending on the model's complexity, each package is hierarchically organized into additional packages and classes. Each class is designed to perform a singular task and strives to be as self-contained as possible. This design not only ensures that each class operates independently but also shields the user from the need to provide inputs or information unrelated to the specific model they are utilizing.

The new packages include:

- `geoet.data` is the package responsible for managing the data and variables of all the ET models. The data are encapsulated in singleton static classes that are simultaneously available to all the other classes of estimating ET, as previously described for the data classes functional to WHETGEO.

- `geoet.inout` package was crafted to facilitate the management of model inputs and outputs. As the model's evolution
aligns with contemporary trends to separate model-agnostic data from the models themselves. This software structure, in fact, allows for the incorporation of diverse data ingestion and extraction methodologies and is pivotal in harnessing novel data handling advancements as they emerge.

Besides the packages dedicated to ancillary shared task, packages were dedicated to contain the various algorithms for each of the evapotranspiration estimation model used:

- Priestley-Taylor formulation (Priestley and Taylor, 1972) of ET is implemented in the `geoet.priestleytaylor` package;

- Penman-Monteith FAO model is implemented in the `geoet.penmanmonteithfao` package;

- Prospero model is more conceptually and computationally complex than the other two models and, as can be seen in Figure 6, is split into various packages and the relative packages are named `geoet.transpiration.*`.

A specialized package has been developed exclusively for estimating the soil evaporation, designated as `geoet.soil-evaporation.solver`. In this phase, the computation of soil evaporation employs the Penman-Monteith method. However, the software's architecture is designed to facilitate effortless integration and development of novel techniques based on the theory supported by the work by Lehmann et al. (2014).

Enclosed within the `geoet.radiation.*` packages lie the procedures inherited from the earlier ETP-GEOframe version,
alongside the methodologies introduced in this study, expounded in the Supplement, which build upon the parameterizations of the processes delineated in de Pury and Farquhar (1997) and Ryu et al. (2011).







**Figure 6.** The GEOframe ET original packages, constituting ETP-GEOframe, (Bottazzi, 2020) (on the left) and the new packages (on the right) containing in GEOET. This refactoring was necessary to accommodate multiple options for estimating the water stress factor, incorporate a more sophisticated radiation budget, and integrate a model for root functioning.



### 4.1.1 How to add a new model?

By following the OOP and generic programming principles of GEOSPACE-1D, integrating a new model into the existing code-base becomes straightforward. To illustrate this, consider the introduction of a new stress model. To incorporate it, a concrete class is created within the `geoet.stressfactor.methods` package, such as `HydroDemo.java`, where the numerical equation for the new method is defined. Furthermore, a solver class is developed within the `geoet.stressfactor.solver` package, for instance, `HydroDemoSolverMain.java`. This class solves the equation by directly invoking methods from the concrete classes. This integration is achieved through addition, requiring no modifications to previously written code.

The newly created class, `HydroDemo.java`, extends the existing abstract class, `WaterStressFactor.java` (see Section 5.4). This new class provides the implementation of the concrete method defined in the abstract class and inherits all of its characteristics. Finally we update the file `WaterStressFactorFactory.java` with the string that users must enter in the model executable to specify the new method they wish to use, namely "HydroDemo".

## 5 About the stress factors models and their deployment

As highlighted in the Supplement, the modeling of stomatal conductance, often referred to as "vegetation stress factor", stands as a central subject of debate within the scientific literature. This contentious issue has spurred the emergence of multiple theoretical frameworks, each trying to provide a comprehensive understanding of this intricate physiological process (Daly et al., 2004).

It is essential to have a computer modelling framework that facilitates the integration of emerging theories without compromising the integrity of the current code. This capability of GEOSPACE-1D ensures that it can adapt in harmony with new research advancements, thereby enhancing the resilience and relevance of its computational framework.

The scientific literature unfolds two primary families of parameterizations governing leaf conductance (Yu et al., 2017), the Jarvis model (Jarvis et al., 1976) and the Ball-Berry one (Dewar, 2002) and it is pivotal to create an informatics infrastructure capable of implementing both these formulations. To achieve this, we constructed an adaptable set of packages: `geoet.stressfactor.*`.

The forthcoming implementation we are detailing must seamlessly function with both the GEOET model independently and when integrated with GEOSPACE-1D. In each scenario, it should have the flexibility to accommodate different stress factor models. This abstraction is essential as stress factor data pertaining to each control volume is not only crucial for GEOET but also necessary for BrokerGEO and WHETGEO within the GEOSPACE-1D framework.

So far, two stress parameterizations were implemented, the so called Jarvis model (Jarvis et al., 1976; Macfarlane et al., 2004; White et al., 1999) and the Medlyn stomatal conductance model (Medlyn et al., 2011).





## 5.1 The Jarvis Model for stresses

As described in the Supplement, the Jarvis model implemented in GEOET follows the version of the model proposed by White
et al. (1999) and by Macfarlane et al. (2004), where the stomatal conductance is equal to:

$$g_s = g_{s,max} \cdot f_R(R_{PAR}) \cdot f_T(T_a) \cdot f_\delta(\delta_a) \cdot f_\psi(\psi_l) \tag{9}$$

where we have stresses for the photosynthetically active radiation (PAR) $f_R(R_{PAR})$, the air temperature $f_T(T_a)$ and the
water pressure deficit $f_\delta(\delta_a)$. All the information about the $f$ functions implemented are detailed in the dedicated Supplement
and are implemented in the `EnvironmentalStress.java` class.

As regards the water stress, despite the wealth of literature, in a considerable number of soil-plant model, water availability
directly depends on $\theta$ (Verhoef and Egea, 2014a) rather than on $\psi$. Following what was done in the precursor GEOframe-ETP
model (Bottazzi, 2020), we maintained the FAO water stress factor $K_s$ (Appendix A) and a new formulation was added as
function $f_\theta(\theta)$, the "Fraction of Transpirable Soil Water" (FTSW), to be used in the Jarvis model (Eq. 9). The formula itself
is a representation of the relationship between soil moisture content ($\theta$), wilting point ($\theta_{WP}$), and field capacity ($\theta_{FC}$), and it
is commonly used to assess plant water stress and to make decisions related to irrigation management and water conservation.
The java class involved in the computation of the FTSW is the `LinearWaterStressFactor.java` class and computes
the water stress factor in each control volume of the grid computational soil according to the following:

$$g_{w,i} = \frac{\theta_i - \theta_{WP,i}}{\theta_{FC,i} - \theta_{WP,i}} \tag{10}$$

where:

- $g_{w,i}$ is the water stress factor in the $i$-th control volume;

- $\theta_i$ represents the current soil water content in the $i$-th control volume;

- $\theta_{WP,i}$ is the wilting point of the $i$-th control volume of the soil, which is the moisture level at which plants can no longer
  extract water from the soil effectively, leading to wilting;

- $\theta_{FC,i}$ represents the field capacity of the $i$-th control volume of the soil, which is the maximum amount of water the soil
  can hold against the force of gravity.

The $\theta_i$ value derived from the WHETGEO model, likewise, the parameters $\theta_{WP,i}$ and $\theta_{FC,i}$ are customized for each control
volume, as it is the user's choice to discretize the soil column and determine the number of layers and associated parameter
values. Consequently, the values of $\theta_{WP,i}$ and $\theta_{FC,i}$ are specific to each individual soil layer and, by extension, to all the
control volumes containing them.

## 5.2 The Medlyn model for stresses

The Medlyn model (Medlyn et al., 2011) is part of the Ball-Berry-Leuning (BBL) family of models (Ball et al., 1987; Leuning,
1990; Dewar, 2002) and it has been modified in various ways since the original paper. For instance, in Dewar (2002) the form





given to it is:

$$g_B = g_0 + g_1 \frac{A_n}{(C_s - \Gamma)\left(1 + \frac{\delta_a}{e_0}\right)} \tag{11}$$

where: $g_B$ is the Ball-Berry-Leuning conductance; $g_0$ is the value of $g_B$ at the light compensation point; $g_1$ and $e_0$ are
empirical coefficients; $A_n$ is the net leaf $CO_2$ assimilation rate; $C_s$ is the $CO_2$ concentration at the leaf surface; $\Gamma$ is the $CO_2$
concentration at the compensation point; and finally $\delta_a$ is the water vapor pressure deficit.

An interesting form of the BBL was obtained in Medlyn et al. (2011), under the hypothesis of optimal photosynthesis theory,
which is:

$$g_B = 1.6\left(1 + \frac{g_1}{\sqrt{\delta_a}}\right)\frac{A_n}{C_s} \tag{12}$$

The equation (12) has been integrated into the GEOSPACE-1D framework simply by adding a class:

- `MedlynStressFactor.java`

and it is fully functional when a time series of net carbon assimilation is provided as input. Its scope actually, is to be a
placeholder for a future OMS3 component dedicated to computing net carbon assimilation within the GEOSPACE-1D.

## 5.3 Estimating the global stress on the basis of local stresses

Given the discretization of soil into multiple layers within GEOSPACE-1D, it becomes imperative to establish a method for
distributing total stress estimates based on local stresses and reciprocally allocating transpiration demands across these soil
layers. The mechanism through which soil moisture is transported into plants, thereby facilitating E$_l$ is primarily via the roots
(Carminati and Javaux, 2020), hence, despite its simplicity, the introduction of a root functioning model is essential. Within
GEOSPACE-1D, root models are integrated into GEOET, through:

- the `geoet.rootdensity.methods` package;

- the `geoet.rootdensity.solver` package.

Eventually, at this point, the information derived by the root models can be used to partition and estimate the total stress
acting on the plants.

GEOET implements three ways to estimate it:

- **Average Water Stress Factor**

  This method is implemented in the `AverageSF.java` class, according to which the representative water stress factor
  value at the $n$-th instant, $G_w$, is given by the arithmetic mean of the stress values characterising the soil column:

$$G_w = \frac{\sum_{i=1}^{N} g_{w,i}}{N} \tag{13}$$

  where $N$ is the total number of control volumes affected by the roots.





**– Size Weighted Water Stress Factor**

This method is implemented in the `SizeWeightedSF.java` class, according to which the representative water stress factor value at the $n$-th instant, $G_w$, is given by the weighted average of the water stress values characterising the soil column as a function of the size $dx$ of the control volumes affected by the roots:

$$G_w = \frac{\sum_{i=1}^{N} g_{w,i} \cdot dx_i}{\sum_{i=1}^{N} dx_i} = \frac{\sum_{i=1}^{N} g_{w,i} \cdot dx_i}{\eta_R} \tag{14}$$

It is specified that the sum of the size $dx_i$ of the control volumes affected by the root system coincides with the depth $\eta_R$ of the roots themselves.

     **– Root Density Weighted Water Stress Factor**

This method is implemented in the `RootDensityWeightedSF.java` class, according to which the representative water stress factor value at the $n$-th instant, $G_w$, is given by the weighted average of the water stress values characterising
the soil column as a function of the root density $\rho_{R,i}$ of the control volumes affected by the roots:

$$G_w = \frac{\sum_{i=1}^{N} g_{w,i} \cdot \rho_{R,i}}{\sum_{i=1}^{N} \rho_{R,i}} = \frac{\sum_{i=1}^{N} g_{w,i} \cdot \rho_{R,i}}{\rho_{Root}} \tag{15}$$

In this case the sum of the root density in each control volumes $\rho_{R,i}$ affected by the root system coincides with the total root density in the root zone $\rho_{Root}$.

## 5.4    Informatics: factory pattern for stress estimator

The multiplicity of parameterizations in the literature for the calculation of the stomatal conductance, as well as the numerous formulations that can be used for the computation of a representative stress factor value, pose precisely the problem of how to implement a structure that is flexible to current and future changes. To do this, we relied on the potential of the design, i.e., the Simple Factory Pattern (Gamma et al., 1995a). If we consider the case of the computation of the representative water stress factor value, we practically need different subclasses that implement several methods with the possibility of the user to choose
one of these classes, and also add new method in the future. Indeed, employing the Simple Factory Pattern (Freeman et al., 2008; Gamma et al., 1995b) enables the encapsulation of object creation logic within the factory class.

     As shown in Figure 7, `RepresentativeSF` class is an abstract class and three java classes that implement the concrete methods to compute the representative water stress factor:

     – `RootDensityWeightedSF`;

– `AverageSF`;





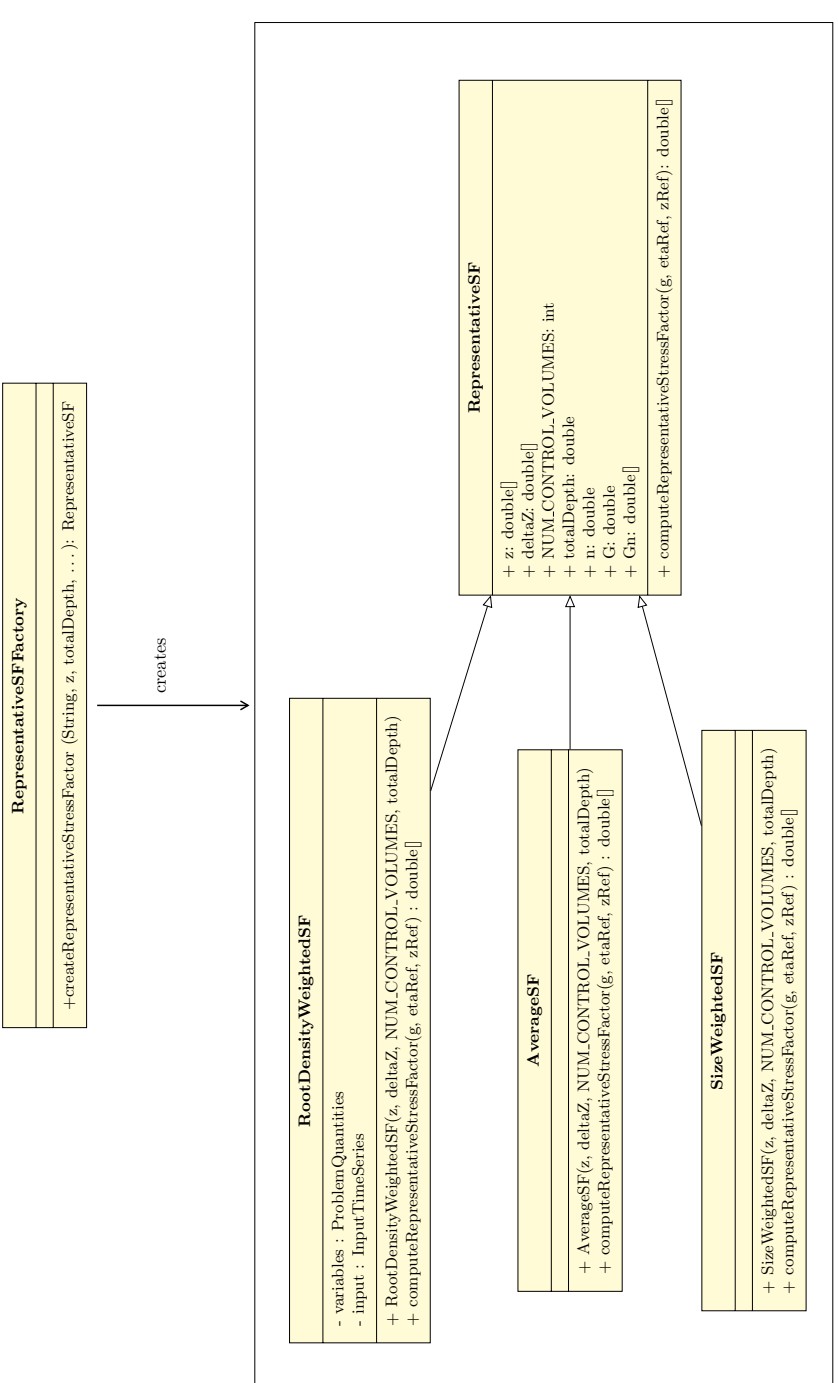

**Figure 7.** UML class diagram for the choice of the representative water stress factor computation model. The `RepresentativeSF` class defines the interface that is implemented by the concrete classes `RootDensityWeightedSF`, `AverageSF` and `SizeWeightedSF`.



– `SizeWeightedSF`.

A closer inspection of Figure 7 reveals that the `RepresentativeSFFactory` class accomplishes the task of implementing one of the concrete classes. In particular the `RepresentativeSFFactory` class is responsible for creating instances of `RepresentativeSF` or its subclasses based on a specified type provided as input. The `RepresentativeSFFactory` class is connected to the classes `RepresentativeSF` and its possible subclasses through the `createRepresentative-StressFactor` method. The factory creates instances of `RepresentativeSF` or its subclasses based on the specified type and returns them as output.

In the context of computing water stress within each control volume, an abstract and factory class framework were developed, like the one just described, using the java classes `WaterStressFactor` and `WaterStressFactorFactory`. This architecture was designed with the foresight of accommodating novel formulations.

## 6 BrokerGEO actions and packages

It is important to note that the ET (or T) flux drawn from the soil from each control volume estimated by GEOET, is subject to a water stress which depends on the the the existing water content estimated by WHETGEO, establishing a feedback between the two software components. To implement the feedback, BrokerGEO distributes the evaporative demand among the soil control volumes.

The three methods implemented below mirror the methods used to obtain the partition of the water stresses factors in GEOET for any time step:

*AverageWaterWeightedMethod*

$$ET_{ref,i} = \frac{g_{w,i}}{\sum_{i=1}^{N} g_{w,i}} ET_{ref} = \frac{g_{w,i}}{N\,G_w} ET_{ref} \tag{16}$$

*SizeWaterWeightedMethod*

$$ET_{ref,i} = \frac{g_{w,i} \cdot dx_i}{\sum_{i=1}^{N} g_{w,i} \cdot dx_i} ET_{ref} = \frac{g_{w,i} \cdot dx_i}{G_w \eta_R} ET_{ref} \tag{17}$$

*RootWaterWeightedMethod*

$$ET_{ref,i} = \frac{g_{w,i} \cdot \rho_{R,i}}{\sum_{i=1}^{N} g_{w,i} \cdot \rho_{R,i}} ET_{ref} \tag{18}$$

where $ET_{ref}$ is the reference flux at the $n$-th instant, $ET_{ref,i}$ is the splitted flux at the $n$-th instant in each $i$-th control volumes, $N$ is the total number of control volumes affected by the roots, $dx$ is the size of the control volumes, $\eta_{ref}$ is the depth at which we want to divide the flow, which can be the depth of the roots or the depth of the soil layer that is considered for evaporation. Finally $\rho_{R,i}$ is the root density in each control volumes and $\rho_{Root}$ is the total root density. The $g_{w,i}$ are the water stress factor in the $i$-th control volume at the $n$-th instant, $G_w$ is the representative water stress factor and the other symbols used have the meanings mentioned above.



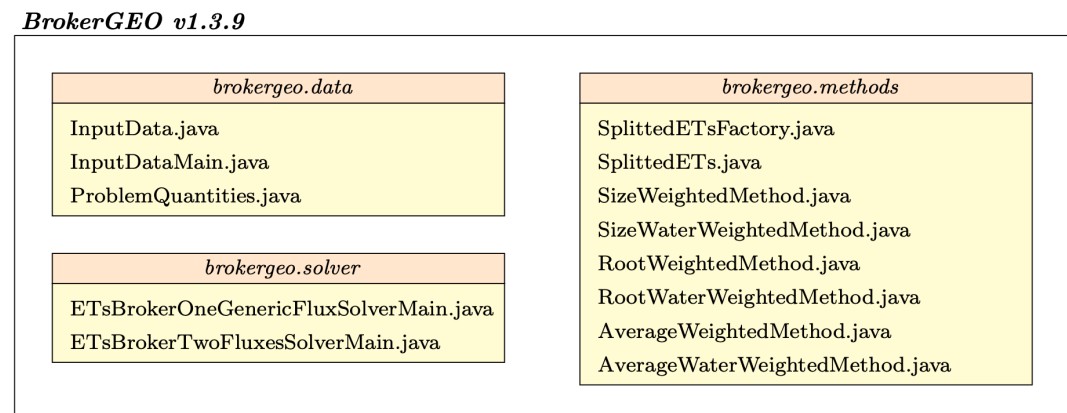

**Figure 8.** IT software structure of BrokerGEO v1.3.9: packages and classes.

## 6.1 BrokerGEO informatics

BrokerGEO is an independent OMS3 component used in GEOSPACE-1D to connect WHETGEO-1D to the four different evapotranspiration models of GEOET. The BrokerGEO 1.3.9 version is mainly composed of three packages shown in Figure 8:

- `brokergeo.data`

- `brokergeo.solver`

- `brokergeo.methods`

The `brokergeo.data` package assumes the role of overseeing the input data and variables associated with the models. It leverages on two classes, namely `ProblemQuantities` and `InputData` for input management. These classes are subsequently linked to the classes responsible for computing the partitioned evapotranspiration within each control volume, which are within the `brokergeo.solver` package.

The `solver` packages contains classes that are designed to invoke methods from the classes in `brokergeo.methods`, enabling the computation of solutions.

In the UML diagram depicted in Figure 9, the package comprises two key solver classes: firstly, `ETsBrokerOneGeneric-FluxSolverMain`, which is tailored for computing only the split evapotranspiration (`ETs`). This approach aligns with the prevalent practices in evapotranspiration modeling literature, where most models estimate the comprehensive evapotranspirative flux, encompassing both vegetation transpiration and bare soil evaporation. However, models like Prospero and TranspirationBudget are dedicated exclusively to transpiration computation. Secondly, `ETsBrokerTwoFluxesSolverMain` is specifically designed to calculate the split transpiration ($E_l$) and split soil evaporation ($E_s$) as distinct components.



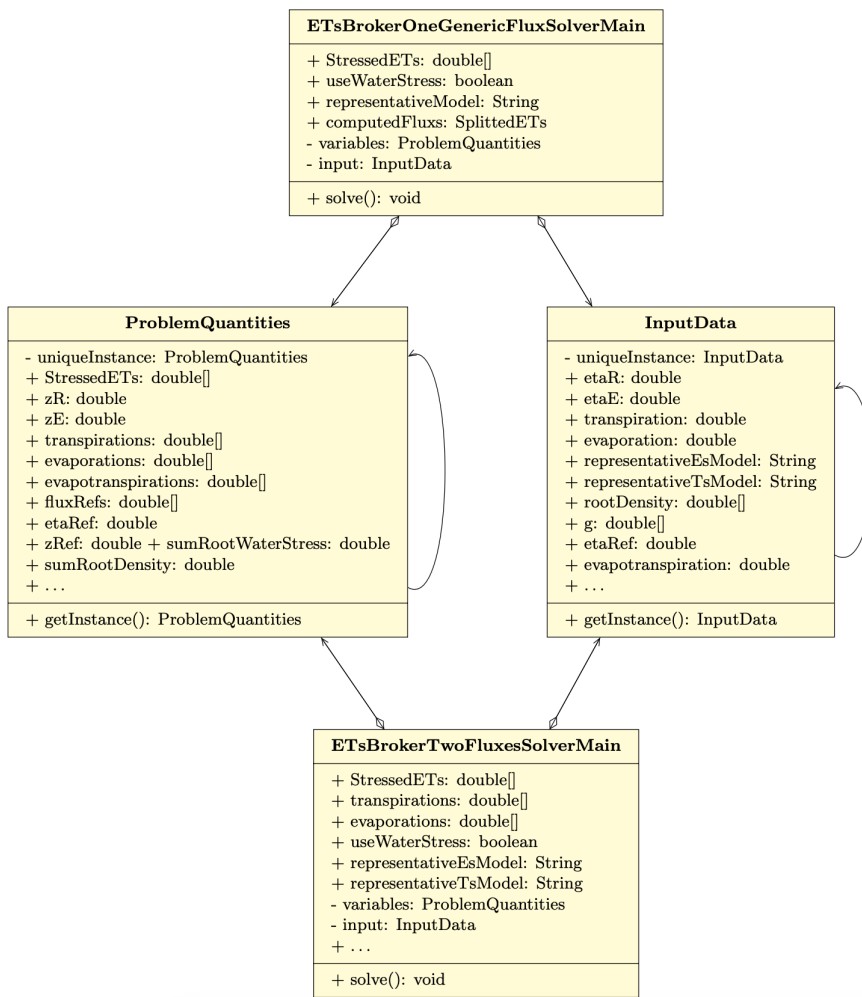

**Figure 9.** UML class diagram for the `ETsBrokerOneGenericFluxSolverMain` class and `ETsBrokerTwoFluxesSolverMain` class. The classes compute the splitted evapotranspiration directly calling the methods of the concrete classes through the Simple Factory Pattern shown in Figure 10. The graph points out the relation of "aggregation" and "association" between the solver classes and `ProblemQuantiites InputData` classes.



Upon a more detailed examination of Figure 9, it becomes apparent that the connection between the solver classes, specifically `ETsBrokerOneGenericFluxSolverMain` and `ETsBrokerTwoFluxesSolverMain`, and the two classes `ProblemQuantities` and `InputData` can be characterized as a hybrid of both "association" and "aggregation."

At the bottom of the programming chain, three classes are responsible for implementing the three equations (16, 17, 18): `AverageWaterWeightedMethod.java`, `SizeWaterWeightedMethod.java` and `RootWaterWeighted-Method.java`. Any other possible partition formula can be implemented as well by simply adding a class in the `method` package.

## 6.2 BrokerGEO Factory Pattern

Given that multiple methods for partitioning evapotranspiration (ET) can be employed, with the potential for introduction of new methods in the future, the most adaptable design choice for the IT architecture of BrokerGEO was the Simple Factory Pattern (Gamma et al., 1995a).

As shown in Figure 10, `SplitterETs` class is the abstract class and it contains only the abstract method *computeStressedETs*. Therefore, we have six possible alternatives that implement a concrete method to compute the splitted flux:

- `AverageWaterWeightedMethod`,

- `AverageWeightedMethod`,

- `RootWaterWeightedMethod`,

- `RootWeightedMethod`,

- `SizeWaterWeightedMethod` and

- `SizeWeightedMethod`.

The connection between the *SplitterETsFactory* class and the *SplitterETs* class is established through the *createEvapoTranspirations* method. This method enables the factory to produce instances of *SplitterETs* and its subclasses based on the specified type string that the user may specify as input to choose which model to use and the corresponding method is returned.

Since BrokerGEO operates as an autonomous OMS3 component, all the implemented numerical methods are capable of partitioning a reference flux. This reference flux may represent simple evaporation, transpiration, or the entire evapotranspiration flux.



**Figure 10.** UML class diagram for the Java Simple Factory applied for the choice of the splitter evapotranspiration method. The `SplittedETs` class defines the interface that is implemented by the concrete classes `AverageWaterWeightedMethod`, `AverageWeightedMethod`, `RootWaterWeightedMethod`, `RootWeightedMethod`, `SizeWaterWeightedMethod` and `SizeWeightedMethod`. The UML graph points out the relation of "aggregation" and "association" between the abstract class `SplittedETs` and `ProblemQuantiites` and `InputData` classes.



**Table 1.** Summary of the main settings for Baseline Simulation: This table presents key parameters and configurations utilized in the baseline simulation.

| | |
|---|---|
| Soil column depth | 2.5 m |
| n° Soil type | 1 |
| n° Control volumes | 250 |
| Time step simulation | hourly |
| Initial condition | Hydrostatic with $\psi = 0$ m at the bottom |
| Top boundary condition | Top Coupled |
| Bottom boundary condition | Free drainage |
| Canopy height | 3.5 m |
| LAI | 2.5 - 4 |
| Root depth | 2 m |
| Root density | Willow "Spike II" experiment |
| Type of stress applied | Water stress |
| $G_{Tw}$ | RootDensityWeightedSF |
| $G_{Ew}$ | AverageSF |
| Transpiration splitter | RootWaterWeightedMethod |
| Evaporation splitter | AverageWaterWeightedMethod |

## 7 Unveiling GEOSPACE-1D capabilities on practical applications

To demonstrate the capabilities of the model, we present a simulation called as "baseline simulation" (BSL) which is described below with the scope to showcase the potential applications and capabilities of the coupled system of GEOSPACE-1D model and verify its correct functioning.

The BSL was done by forcing the model with the input data of the two-month dataset from the "Spike II" experiment whose forcings are shown in Figure 11. A comprehensive description of the experiment is reported in Queloz et al. (2015), Benettin
et al. (2019), Benettin et al. (2021a), Benettin et al. (2021b) and Nehemy et al. (2021).

The "Spike II" BSL can be described as a one-dimensional simulation of a homogeneous column of soil of 2.5 m depth with a plant of height of 3.5 m, i.e. a willow. The soil column was discretized with a uniform grid space of 250 control volumes and we carried on a hourly time-step simulation.

The soil hydraulic properties are described with the Van Genuchten's model (Table 2), and most of them are taken from the
work of Asadollahi et al. (2022). The initial condition is assumed to be hydrostatic with $\psi = 0$ m at the bottom.

Table 1 provides a comprehensive overview of the settings employed in the baseline simulation.

The surface boundary condition is the rainfall and irrigation shown in Figure 11 (a). WHETGEO-1D includes a surface water coupled model which allows the formation of surface ponding (Tubini and Rigon, 2022), and so the surface boundary condition may change from the Dirichlet type - prescribed water suction - to the Neumann type - prescribed flux - and vice-



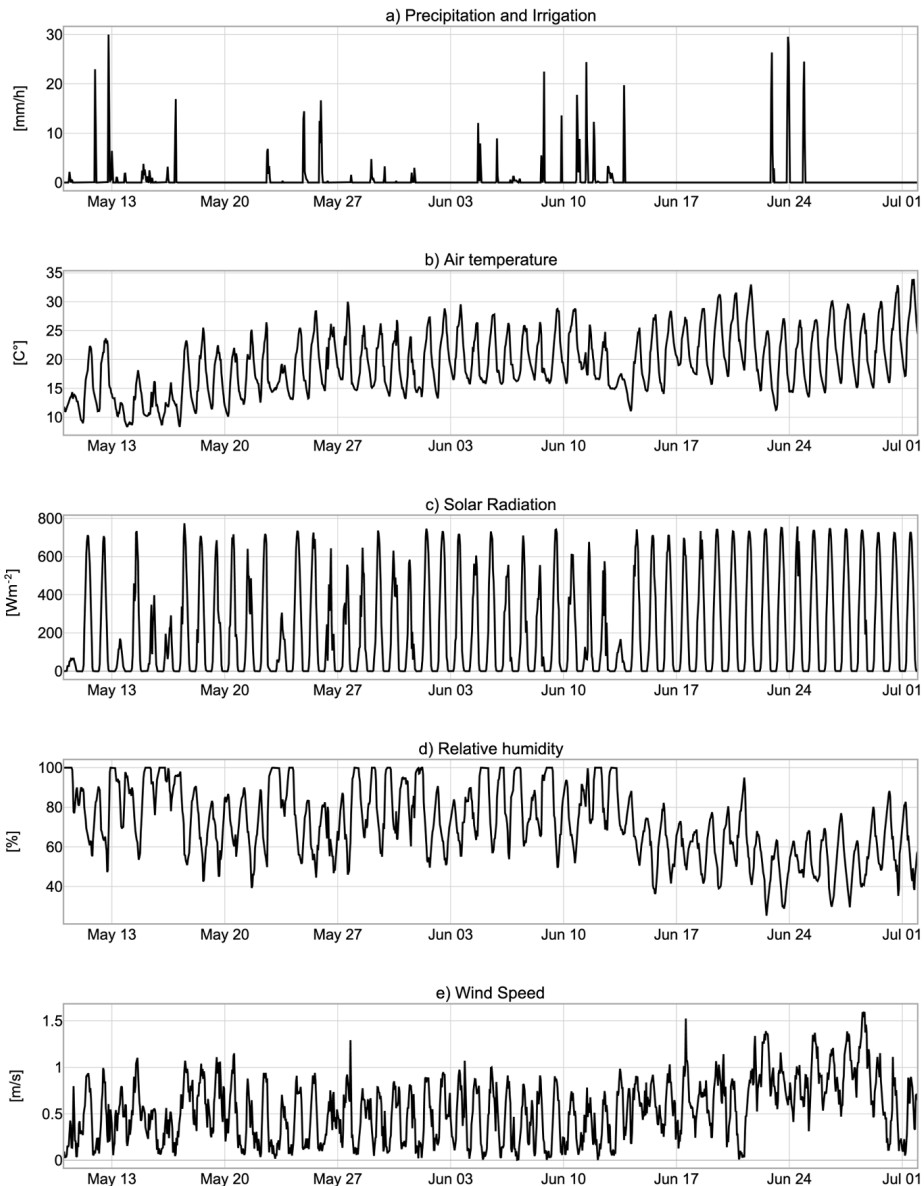

**Figure 11.** Timeseries of the dataset from "Spike II" experiment provided by the meteo station (MeteoMADD, MADD Technologies Sàrl, Switzerland). The data has been accurately aggregated on an hourly scale to be used as input of the simulation with GEOSPACE-1D model.

**Table 2.** Hydraulic properties of the soil column for the baseline simulation.

| $\theta_r$ [-] | $\theta_s$ [-] | $\theta_{WP}$ [-] | $\theta_{FC}$ [-] | $\alpha$ [m$^{-1}$] | $n$ [-] | $K_s$ [m s$^{-1}$] |
|---|---|---|---|---|---|---|
| 0.05 | 0.3 | 0.0803 | 0.205 | 6 | 1.31 | $3.4722 \times 10^{-5}$ |





versa, according to the occurring process. At the bottom of the column, we prescribed a free drainage boundary condition with
a gravity-driven transient.

For the evapotranspiration model, we utilized the GEOET-Prospero model, driven by the input data shown in Figure 11. Our
analysis focused on a plant of height of 3.5 m, with leaf area index (LAI) linearly varying from 2.5 to 4 at the beginning and
end of the simulation, respectively. As root depth and density parameters we used the "Spike II" experiment measurements of
the willow and they were kept constant throughout the simulation.

Finally, among the Jarvis stress only the water stress is considered, excluding all environmental stresses and the represen-
tative water stress factor for transpiration $G_{Tw}$ is computed with the root density weighted method. The representative water
stress factor for evaporation $G_{Ew}$ is calculated as the arithmetic mean of the stress values characterizing the depth of the
evaporation layer, located 0.2 m from the soil surface.

The partitioning of evaporation and transpiration between the control volumes is done by BrokerGEO using `AverageWater-`
`WeightedMethod` for evaporation, and the `RootWaterWeightedMethod` for transpiration.

For further details, please refer to the dedicated folder available on the Zenodo repository (D'Amato and Rigon, 2024) and
the original Thesis (D'Amato, 2024).

Figure 12 depicts the temporal evolution of soil water potential following the rainfall event illustrated in Figure 11 . Darker
blue patterns indicate higher soil water potential, demonstrating a noticeable increase following irrigation or rainfall episodes.
In the last events, saturation occurs, leading to surface ponding, eventually resulting in the formation of an infiltrating saturated
front. The above situation is better clarified in Figure 13 where saturation is evident to be generated at the surface (blue line)
and quickly propagate in the soil down to -1.5 m.

As depicted in Figure 14, the GEOSPACE-1D outputs provide the necessary data to visualize water pressure and water con-
tent profiles at different time steps. This particular figure represents the scenario estimating the combined effects of infiltration
and transpiration on soil moisture. In contrast, Figure 15 serves as a visual comparison, illustrating the dynamics of infiltration
without water extraction by the plant. In this scenario, we replicated the baseline simulation while excluding the evapotran-
spiration process, thus following the development of the infiltration process in the soil column without evapotranspiration. It's
evident that the soil column becomes more saturated compared to the baseline simulation. The mass balance closure is usually
verified in the simulation and, in this case, the error in the closure is around $10^{-9}$ m as shown in the supplemental material.

The differences between the two cases of infiltration are particularly apparent when comparing Figures 14 and 15. For a more
comprehensive analysis, it is recommended to compare Figure 12 and 16. The total bottom flux without evapotranspiration
amounts to 602.31 mm, a stark contrast to the 180.09 mm observed in the baseline simulation, resulting in a 234.6% increase
in groundwater recharge.

Figure 17 represents the evolution of the water stress in time and depth in the BSL. As said before, the root depth is
considered constant in time but a time series describing the root depth growth could be provided as an input to the model, if
available. The bulk stress factor can be easily estimated from the data provided and is shown in the Zenodo folder material
(D'Amato and Rigon, 2024).



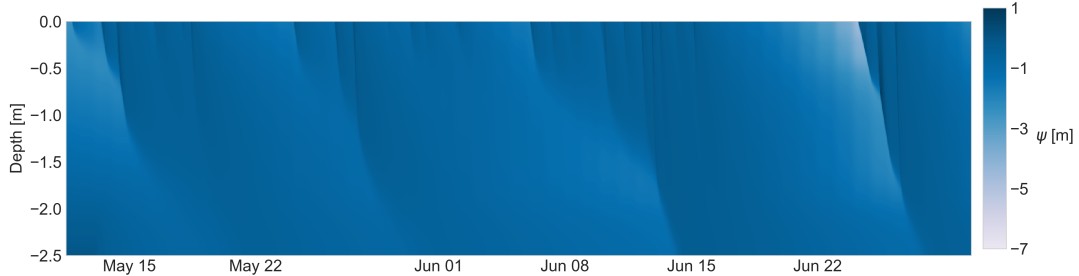

**Figure 12.** Soil water potential behaviour in the baseline simulation. The plot displays depth in meters, with a color scale representing soil water potential ($\psi$) along the depth.

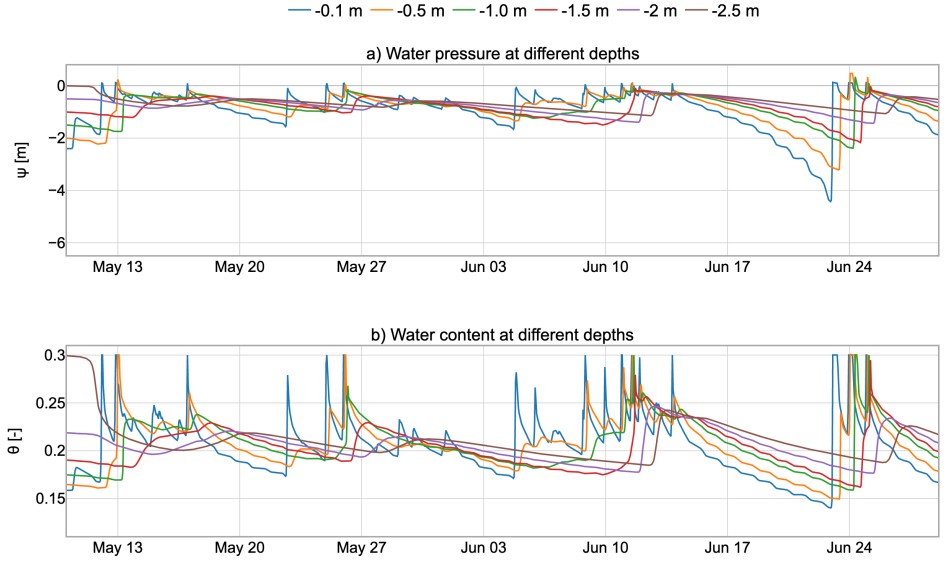

**Figure 13.** (a) Soil water potential, and (b) water content evolution with time at different depth in the baseline simulation.

Solar radiation is measured in this experiment, but it undergoes filtration as it passes through the plant canopy, as described
by the model presented in the Supplement. The remaining radiation, known as residual radiation, is depicted in Figure 18, and it reaches the soil, where it is utilized to estimate soil evaporation using the PM-FAO model. The estimated soil evaporation and transpiration rates are then shown in Figure 19.





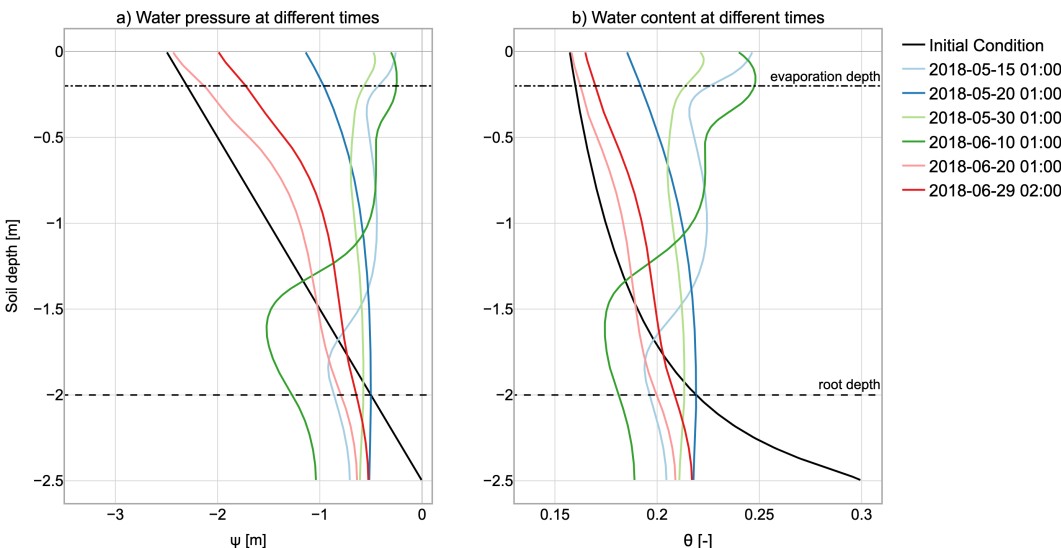

**Figure 14.** (a) Evolution of soil water potential, and (b) water content along the soil profile at various arbitrarily chosen times in the baseline simulation.

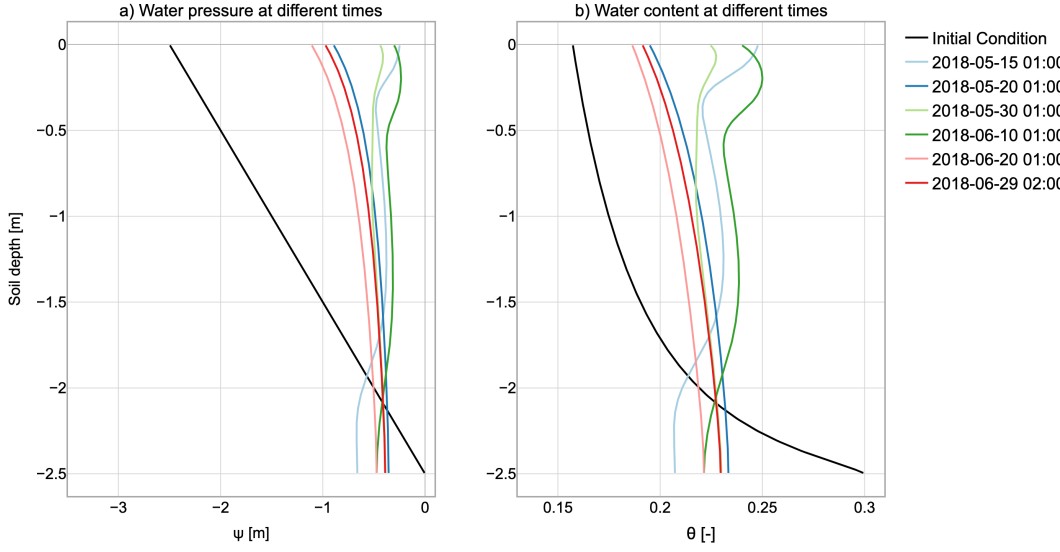

**Figure 15.** (a) Evolution of soil water potential, and (b) water content along the soil profile at various arbitrarily chosen times in a scenario without Evapotranspiration.





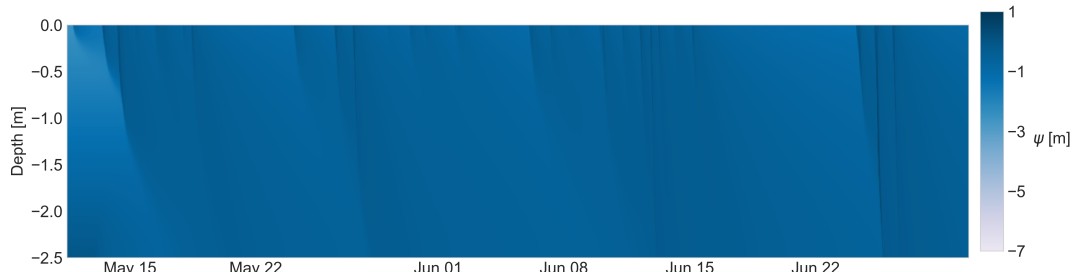

**Figure 16.** Soil water potential behaviour in scenario without Evapotranspiration. The plot displays depth in meters, with a color scale representing soil water potential ($\psi$) along the depth.

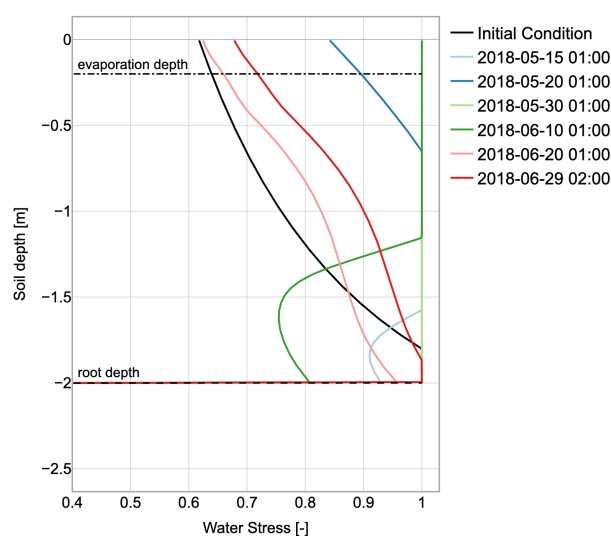

**Figure 17.** Evolution of water stress factor along the soil profile at various arbitrarily chosen times in the baseline simulation.

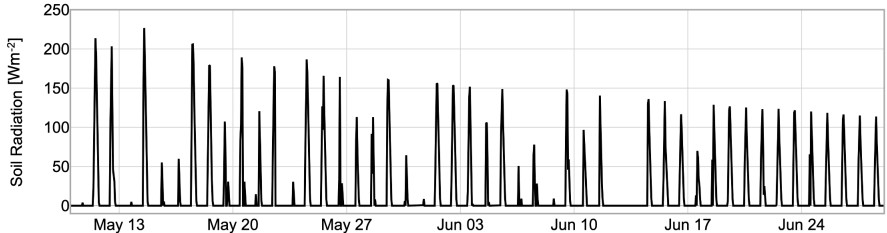

**Figure 18.** Radiation reaching the ground, given the prescribed LAI as input, in the baseline simulation.





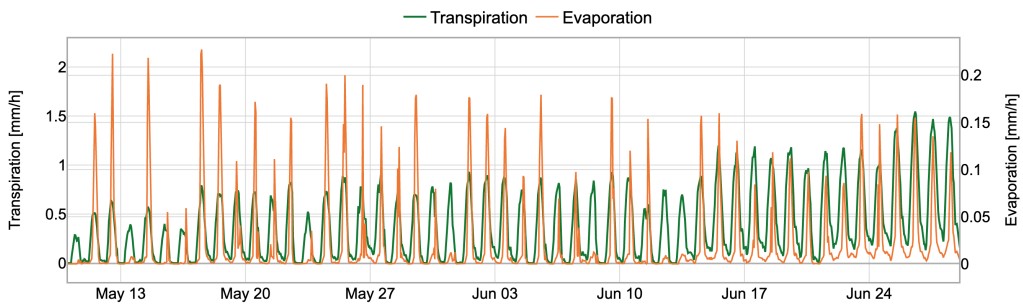

**Figure 19.** Evolution of Transpiration (green) and Evaporation (orange) fluxes with time in the baseline simulation.





## 7.1 User information: Input and Output

The GEOSPACE-1D system exhibits a modular structure, necessitating various inputs contingent upon the specific modules
employed. Table 3 summarizes the general inputs required for all components in GEOSPACE-1D. These simulation parameters,
including the simulation's temporal parameters, are user-defined in the OMS3 .sim file.

In the context of WHETGEO, input data can be categorized into two main key components: time series and computational
grid data (Table 4). Time series data primarily serve to define the boundary conditions for the case study. These time series are
encapsulated within .csv files specific for OMS3 format. Computational grid data encompasses domain discretization, initial
conditions, and terrain parameters, all stored in Network Common Data Form (netCDF) files. The processing of time series
and computational grid data is executed using specialized Python modules integrated into the `geoframepy` package (Tubini
and Rigon, 2021).

Specifically, the time series data for boundary conditions needs meteorological information about precipitation, the presence
of surface ponding, and the hydraulic condition at the base of the soil column. When addressing computational grid considera-
tions, it is essential to specify the structure of the soil column, such as the soil horizons and hydrological/hydraulic parameters,
or to employ literature data for saturated hydraulic conductivity, water content at saturation, residual values, and the water
retention curve parameters (SWRC), as outlined in WHETGEO (Tubini, 2021). Most of these soil parameter, implemented as
default in WHETGEO, are available in Bonan (2019), but any other dataset can be used as well by changing the appropriate
configuration files.

The input data required by GEOET are time series of meteorological forcings, that could be downloaded from online weather
service portals. Specifically, the input data includes time series of air temperature, wind speed, relative humidity, solar radiation,
and pressure, but they depend on the evapotranspiration model used from those available (Table 5). In the context of vegetation
characterization, Leaf Area Index data and canopy height, acquired from literature or satellite sources, are fundamental for
the computation of transpiration flux. Root depth ($z_R$) and density ($\delta_R$) have to be specified in the creation of computational
grid data , because their information is needed in each control volume. Additionally, parameters associated with stress factor
calculations, such as the water content at wilting point ($\theta_{WP}$) and field capacity ($\theta_{FC}$), can be readily sourced from available
literature. Finally, the user has to specify which model to use to compute the representative water stress factor $G_w$ (Section 5)
and accordingly, also the model to split evapotranspiration with BrokerGEO (Table 6) as described in Section 6.2. The output
data of GEOSPACE-1D depends on the model combination that we use. Table 7 summarizes the GEOSPACE-1D outputs.

The WHETGEO model generates output data in NetCDF format. Nevertheless, these outputs (as well as the inputs) are man-
aged separately by specifically designed OMS3 components, which handle the writing and reading processes. This separation
ensures modularity and flexibility, allowing alternative writers and readers to be provided to adapt to different file formats as
required.

For ease of reading, we provide users with extensively commented Python Notebooks, through which all possible outputs
of WHETGEO-1D can be accessed. Outputs include water suction [m], water content [-], Darcy velocity [m/s], mass balance
error, and others. If the heat transport model is used, the soil temperature is also outputted.





**Table 3.** General inputs required for all components in GEOSPACE-1D. These simulation parameters, including the simulation's temporal parameters, are user-defined in the OMS3 .sim file.

| Temporal Resolution (Minute/Hourly/Daily) | Start-End Date | Point Geographical Location (Long-Lat-Elevation) |
|---|---|---|

**Table 4.** Input data for WHETGEO component, categorized into two main elements: time series and computational grid data.

WHETGEO

| Time Series | Top BC |
|---|---|
| | Bottom BC |
| | Precipitation |
| Computational grid data | Initial condition ($\psi_0$, $T_0$, $z_R$, $\delta_R$) |
| | Soil discretization (depth per each layer) |
| | Soil parameters ($\theta_S$, $\theta_R$, $\theta_{WP}$, $\theta_{FC}$, $K_s$) |
| | Soil parameters for SWRC |

To date, GEOET results have been stored in OMS3 CSV files, and the type of outputs depends on the evapotranspiration model used in GEOSPACE-1D, as highlighted in Table 7. Specifically, if we consider Prospero for transpiration and Penman-Monteith for soil evaporation, the estimated output quantities are: soil evaporation [mm/time], transpiration [mm/time], latent and sensible heat flux [Wm$^{-2}$], leaf temperature [°C], water vapor deficit [-], and so on. For a complete description, please refer to the Jupyter Notebook 00_Notebook of GEOSPACE-1D in the GitHub page model.

It's important to highlight that a common challenge in conducting simulations is the scarcity of data required to perform them. For instance, obtaining data on root density can be difficult, and it's not always guaranteed that a study site will have a meteo station providing all the necessary data outlined in Tables 4 and 5. This is why GEOET offers both simple and complex evapotranspiration models. In cases where all the required data for complex models isn't available, simpler models requiring fewer inputs can be utilized.

Taking again Prospero as an example, it's possible to encounter situations where only a few of the necessary inputs are missing. In such instances, default data from the model or data from existing literature can be used. In the context of evapotranspiration models, radiation is a critical input that must never be absent, as it serves as the primary driver. Similarly, when it comes to roots, while the depth of roots may be known, their density is often unknown. In such cases, one may opt for an arbitrary distribution or reconstruct it using data from literature or more users can use one of the root density growth methods implemented in GEOET. For more information please refer to D'Amato (2024).





**Table 5.** Time series input data for GEOET component. Comparison of input requirements for different evapotranspiration models (Prospero, PM-FAO, and PT) in GEOET

<table>
<tr><td colspan="4" align="center">GEOET</td></tr>
<tr><th>Input</th><th>Prospero</th><th>PM-FAO</th><th>PT</th></tr>
<tr><td>Air Temperature [°C]</td><td>✓</td><td>✓</td><td>✓</td></tr>
<tr><td>Wind Velocity [m/s]</td><td>✓</td><td>✓</td><td>-</td></tr>
<tr><td>Relative Humidity [-]</td><td>✓</td><td>✓</td><td>-</td></tr>
<tr><td>Net Radiation [Wm$^{-2}$]</td><td>✓</td><td>✓</td><td>✓</td></tr>
<tr><td>Short Wave Direct [Wm$^{-2}$]</td><td>✓</td><td>-</td><td>-</td></tr>
<tr><td>Short Wave Diffuse[Wm$^{-2}$]</td><td>✓</td><td>-</td><td>-</td></tr>
<tr><td>Soil Heat Flux [Wm$^{-2}$]</td><td>✓</td><td>✓</td><td>✓</td></tr>
<tr><td>Atmospheric Pressure [Pa]</td><td>✓</td><td>✓</td><td>✓</td></tr>
<tr><td>Leaf Area Index [-]</td><td>✓</td><td>-</td><td>-</td></tr>
<tr><td>Canopy height [m]</td><td>✓</td><td>✓</td><td>-</td></tr>
<tr><td>Model to compute $G_{Tw}$ (Section 5)</td><td>✓</td><td>✓</td><td>✓</td></tr>
</table>

**Table 6.** Overview of key inputs within the BrokerGEO Model: the algorithm for computing reference evapotranspiration ($ET_{ref,i}$), details about the computational grid, and the reference flux, evaporation, transpiration or evapotranspiration, to be splitted.

<table>
<tr><td align="center">BrokerGEO</td></tr>
<tr><td align="center">Model to compute $ET_{ref,i}$ (Section 6.2)</td></tr>
<tr><td align="center">Computational grid information</td></tr>
<tr><td align="center">Reference flux to be splitted (ET)</td></tr>
</table>





**Table 7.** GEOSPACE-1D model Outputs: Hydrological and meteorological variables between WHETGEO Output (.netcdf) and GEOET Output (.csv).

| **WHETGEO Output (.nc)** | *Water Suction [m] |
| --- | --- |
| | *Water content [-] |
| | *Darcy velocity [m/s] |
| | Volume error [m] |
| | Surface runoff [m/s] |
| | *Evapotranspiration [m] |
| | *Water stress [-] |

| **GEOET Output (.csv)** | **Prospero + PM** | **PM-FAO** | **PT** |
| --- | --- | --- | --- |
| Latent Heat Sun [$Wm^{-2}$] | ✓ | - | - |
| Latent Heat Shadow [$Wm^{-2}$] | ✓ | - | - |
| Transpiration [mm/time] | ✓ | - | - |
| EvapoTranspiration [mm/time] | **✓ | ✓ | ✓ |
| Soil Evaporation [mm/time] | **✓ | - | - |
| Leaf Temperature Sun [K] | ✓ | - | - |
| Leaf Temperature Shadow [K] | ✓ | - | - |
| Sensible Heat Sunlit [$Wm^{-2}$] | ✓ | - | - |
| Sensible Heat Shadow [$Wm^{-2}$] | ✓ | - | - |
| Radiation Soil [$Wm^{-2}$] | ✓ | - | - |
| Radiation Sun [$Wm^{-2}$] | ✓ | - | - |
| Radiation Shadow [$Wm^{-2}$] | ✓ | - | - |
| Canopy [-] | ✓ | - | - |
| VPD [-] | ✓ | - | - |

* This variable contains all the output of the current time step

and for each control volumes.

** When we use Prospero, we compute evaporation with PM model

defining as input an evaporation depth for the soil.



## 8   User information: Code availability

The latest executable code of only WHETGEO model with the new classes mentioned before can be downloaded from
geoframecomponents/WHETGEO-1D and can be compiled by following the instructions therein. The version of the OMS3
compiled project can be found here.

The Object Modeling System v.3 (OMS3) is a component-based environmental modeling framework introduced by David
et al. (2013). More information in the Supplement.

While the majority of the content presented thus far holds broad relevance, it's important to note that the deployment example
showcased here pertains to a one-dimensional (1D) context. Comprehensive information on GEOSPACE-1D, intended for both
users and developers, can be found in the supplementary materials. Included within is a Jupyter Notebook labeled with the
prefix "00_", affectionately referred to as "Notebook Zero," which serves as a comprehensive guide for executing the code
pertaining to any of the components. The latest executable code can be downloaded from geoframecomponents/GEOSPACE-
1D and can be compiled by following the instructions therein. Finally, the version of the OMS3 compiled project can be found
here.

The code can be executed in the OMS3 console, which can be downloaded and installed according to the instructions given
at here.

The various components of GEOSPACE v.1.2.9 are finally compiled and grouped in the following `.jar` files:

–   whetgeo1d-1.2.9 includes the WHETGEO model components;

–   brokergeo-1.3.9 includes the BrokerGEO coupler;

–   geoet-1.5.9 includes the GEOET model containing the ET modules, besides the root functioning, the modules that split
the radiation into the canopy and soil layers, the stress factor estimators;

–   buffer-1.1.9 contains ancillary modules for the management of data in memory and their eventual printing;

–   closureequation-1.1.9 manage the equations that are actually solved in WHETGEO;

–   netcdf-1.1.9 manage the input and output from and to netcdf files;

–   numerical-1.0.2 contains the core algorithms for the solutions of the equations.

The integration of all components took place within the Object Modeling System v3 framework (OMS3) and interlinked
within GEOSPACE. Notably, within the aforementioned components, specific to WHETGEO, are modules such as *buffer*,
*closureequation*, *netcdf* and *numerical*. For comprehensive information on these WHETGEO specific components, you can
refer to Tubini (2021). Due to the modularity of the system, whilst the components were developed and can be enhanced
independently, they can be seamlessly used at run time by connecting them with the OMS3 DSL language based on Groovy.
OMS3 provides the basic services and, among them, tools for calibration and implicit parallelization of component runs.

Finally more comprehensive information about GEOframe is available at:





- https://abouthydrology.blogspot.com/2015/03/jgrass-newage-essentials.html

– https://geoframe.blogspot.com/2020/01/gsw2020-photos-and-material.html

## 9 Conclusions

This paper presents and discusses the implementation of the GEOSPACE framework, in particular its one-dimensional development, which stands out among various software for its component-based organization leveraging the OMS3 framework. GEOSPACE-1D comprises three main components: WHETGEO, GEOET, and BrokerGEO. WHETGEO serves as a solver for
the Richards-Richardson equation, employing a novel integration algorithm, incorporating the soil energy budget, and featuring automatic switching between saturated and unsaturated conditions. GEOET implements various evapotranspiration formulas and the Prospero model, which concurrently solves transpiration and heat transport alongside the stationary energy budget. Additionally, GEOET provides information on the temperature and water pressure differentials between leaves and air. BrokerGEO acts as a connector between the GEOET and WHETGEO solvers, redistributing evaporation and transpiration demands
within the soil column. Additionally, the paper delves into the underlying philosophy of the GEOSPACE framework, which can be succinctly summarized as a structure designed to facilitate modifications aimed at exploring soil-plant-atmosphere interactions under various hypotheses and incorporating new research findings. This objective is accomplished by implementing core functionalities as relationships among abstract objects, specifically interfaces, which are subsequently instantiated into concrete classes, allowing flexibility for user/programmer choices. The instantiation mechanism for these classes is achieved using the
factory pattern. A consequence of the above choices is that, for instance, the creation of a new method for estimating the water stress consists in just adding a new class that extends the general interface. Adding a new method for estimating transpiration or evaporation can require a few classes, depending on the complexity of the method, but always obeying the prescription that the code remain "open to extensions and close to modifications". In the cases already deployed, the Priestley Taylor formulas is implemented by using three classes, while the Prospero model requires a few more as shown in Figure 6. Departing from the
strict object-oriented paradigm, which necessitates the creation and destruction of immutable classes, GEOSPACE-1D adopts a more conventional approach, particularly familiar to scientists, for classes dedicated to data containment. Data classes are structured as singletons, instantiated once and continuously updated with new content at each time step.

The model's capabilities are showcased through a "baseline" simulation, which compares infiltration estimates with and without evaporation and transpiration based on data collected in the "Spike II" experiment at the Ecole Polytechnique Fédérale
de Lausanne. The simulations conducted aimed to highlight the functionalities and capabilities of the system, demonstrating its ability to maintain water mass and energy conservation. It's worth noting that during these simulations, the soil column experiences saturation due to a perched water table while transpiration occurs. To provide a basis for comparison, a reference simulation, driven by the same meteorological data, is performed to illustrate the significant difference in soil moisture evolution. Throughout the simulations, the mass budget closure is rigorously monitored and consistently achieved with high
precision, despite the diverse range of simulated conditions. These simulations exhibit a strong correspondence with measured data, as further elaborated in a companion paper currently in preparation.



As part of the project, a collection of companion Jupyter Notebooks has been developed to elucidate the process of inputting data required by GEOSPACE-1D, across various configurations. This aligns with the model's implementation philosophy, which separates the handling of model-agnostic components (written in Python and potentially compatible with other modeling platforms) from the model-specific elements (programmed in Java, in our case). The outcome of any simulation setup is encapsulated in OMS3 ".sim" files, facilitating easy sharing. In line with promoting FAIR principles in modeling, all aspects related to GEOSPACE-1D including executables, Notebooks, simulations, input and output data files are consolidated within OMS3 projects and can be readily distributed. Everything utilized in this paper is included in the dedicated folder available on the Zenodo repository (D'Amato and Rigon, 2024) for third-party inspection and self-instruction. Furthermore, the source code is openly accessible and provided on GitHub, as detailed in the relevant sections of the paper. Additionally, instructional videos demonstrating the system's functionality can be found on the GEOframe blog.

*Code availability.* The software is available on the GEOframe Components GitHub repository, as detailed in Section 8. The source code, written in Java using an object-oriented programming approach, is specifically hosted on GitHub here. Additionally, the corresponding OMS3 project is accessible here. Details about external dependencies are provided in the README file of the GEOSPACE-1D GitHub page.

To execute the code, you must use the OMS3 Console. Instructions for setting up the environment can be found in the README file within the repository. After installing OMS3, follow the guidelines in the Documentation folder, which contains all necessary details about simulation inputs and parameters.

*Data availability.* Data used in this paper is available as part of the "Spike II" tracer experiment dataset (Nehemy et al., 2020), under Creative Commons Attribution (CC BY) license, at doi.org/10.5281/zenodo.4037240. The simulation presented in this work are available on Zenodo 10.5281/zenodo.14269885.

*Video supplement.* Video illustrating the use of the GEOSPACE-1D parts are present in the material of the GEOframe 2022 Summer School

## Appendix A: GEOET

The evapotranspiration component, GEOET, is a deep refactoring of the GEOframe-ETP model (Bottazzi, 2020; Bottazzi et al., 2021). The GEOET framework incorporates four evapotranspiration models, as illustrated in Figure 6. Starting with the simplest model, we have the Priestley-Taylor model (PT), the Penman-Monteith FAO (PM) approximation, an adaptation of the Penman-Monteith model, the Prospero model (Bottazzi, 2020; Bottazzi et al., 2021) and the TranspirationBudget model, a complex computation of the energy budget at the canopy scale described in Rigon and D'Amato (2024). The Priestley-Taylor model was already fully described in the main text, while the PM model and the Prospero model requires some further explanations.





## A1 Penman-Monteith FAO additional information

Chapter 2 of FAO evapotranspiration book (Food and Agriculture Organization of the United Nations) introduces the user to the need to standardize one method to compute reference evapotranspiration ($ET_0$). As described in Bottazzi (2020), the Penman-Monteith FAO is the approximation for the Penman-Monteith, defined for a reference crop as a hypothetical crop with an assumed height of 0.12 m, having a surface resistance of 70 s m$^{-1}$ and an albedo of 0.23. It is widely used especially in agricultural field.

Starting from the Penman-Monteith equation, the FAO approximation for a grass reference surface lead to computation of a reference evapotranspiration as follow:

$$ET_0 = \frac{1}{\lambda} \frac{0.408\Delta(R_n - G) + \gamma\frac{900}{T+273}u_2(e_s - e_a)}{\Delta + \gamma(1 + 0.34u_2)} \tag{A1}$$

where: $ET_0$ is the reference evapotranspiration [mm day$^{-1}$], $R_n$ is the net radiation at the crop surface [MJ m$^{-2}$ day$^{-1}$], $G$ is the soil heat flux density [MJ m$^{-2}$ day$^{-1}$], $T$ is the mean daily air temperature at 2 m height [°C], $u_2$ is the wind speed at 2 m height [m s$^{-1}$], $e_s$ is the saturation vapour pressure [kPa], $e_a$ is the actual vapour pressure [kPa], $(e_s - e_a)$ is the saturation vapour pressure deficit [kPa], and $D$ is the slope vapour pressure curve [kPa °C$^{-1}$],

Using the latent heat constant $\lambda$, the reference evapotranspiration can be converted to the reference latent heat:

$$E_0 = ET_0 \cdot \lambda \tag{A2}$$

FAO computes the actual evapotranspiration using the water stress coefficient $K_s$ and the single crop coefficient $K_c$:

$$E_{FAO} = ET_0 \cdot K_s \cdot K_c \tag{A3}$$

Values for $K_c$ are given by FAO in Table 12 of Chapter 6. The water stress coefficient $K_s$ can by computed as:

$$K_s = \frac{TAW - D_r}{TAW - RAW} = \frac{TAW - D_r}{(1 - p)TAW} \tag{A4}$$

$$RAW = p \cdot TAW \tag{A5}$$

$$TAW = 1000(\theta_{FC} - \theta_{WP}) \cdot Z_r \tag{A6}$$

$$TAW = 1000(\theta_{FC} - \theta) \cdot Z_r \tag{A7}$$

where:



- $K_s$ is a dimensionless transpiration reduction factor dependent on available soil water [0 - 1];

- $D_r$ root zone depletion [mm];

- $TAW$ total available soil water in the root zone [mm];

- $p$ fraction of $TAW$ that a crop can extract from the root zone without suffering water stress [-];

- $\theta_{FC}$ the water content at field capacity [m$^3$m$^{-3}$];

- $\theta_{WP}$ the water content at wilting point [m$^3$m$^{-3}$];

- $\theta_{WP}$ the measured water content [m$^3$m$^{-3}$];

- $Z_r$ the rooting depth [m].

## A2  Prospero model

The Prospero model, Bottazzi (2020) and Bottazzi et al. (2021), is a physically based approach for calculating canopies transpiration ($T$) and soil evaporation $E$. These processes are considered for both sunlit and shaded fractions of the canopy. E from the soil is determined according the FAO Penman–Monteith model meanwhile, T is computed using a modified version of the Schymanski and Or (Schymanski and Or, 2017) (SO) model, which has been upscaled to address canopy-level transpiration and modified to ensure mass conservation during periods of water stress. SO model solves the stationary energy budget coupled with the water vapor transport Dalton's equation and the sensible heat $S$ equation by using a suitable simplification of the Clausius-Clapeyron formula.

For a more in-depth exploration of this topic, readers are encouraged to refer to the comprehensive information provided in Bottazzi (2020) or to Rigon and D'Amato (2024).

To expand the applicability of this equation to the canopy, Bottazzi (2020) chose to implement a two-big-leaf approach (Dai et al., 2004; Luo et al., 2018), utilizing the Sun/Shade model introduced by (de Pury and Farquhar, 1997) and illustrated in the Supplement. This approach enables the calculation of the fraction of the canopy exposed to direct sunlit and the fraction in shade, along with the radiation absorbed by various canopy layers. Specifically, Prospero employs this two-leaf sun-shade approach, while treating the soil as an additional layer. Other environmental variables such as air temperature, relative humidity, wind, and longwave radiation are assumed to remain constant within the canopy, allowing it to be treated as a single large leaf emitting latent heat proportionate to its area and shortwave radiation.

The SO approach overcomes several limitations of the Penman-Monteith equation, particularly in its representation of transpiring leaf area and leaf thermal capacity, along with their feedback on the energy balance. Incorrectly representing the transpiring leaf area can significantly impact the overall energy balance. Consequently, the energy budget can be reformulated in terms of the area capable of exchanging fluxes and the equilibrium leaf temperature as follows:

$$R_s = a_{sH} A_{tr} R_{ll}(T_l) + a_{sE} A_{tr} E_l(T_l) + a_{sH} A_{tr} H_l(T_l) \tag{A8}$$





where

- $a_{sH}$ is the side of the surface exchanging sensible heat (1 for soil, 2 for leaves) [-],

- $a_{sE}$ is the side of the surface exchanging latent heat (1 for amphistomatous leaves) [-],

- $A_{tr}$ is the area exchanging fluxes (radiation, sensible and latent heat) [$\text{m}^2\text{m}^{-2}$],

- $T_l$ is the equilibrium leaf temperature [K].

where the longwave in computed as

$$R_{ll} = a_{sH} A_{tr} \epsilon_l \sigma (T_l^4 - T_a^4) \tag{A9}$$

and T is computed as

$$E_l = c_E(a_{sE}, A_{tr}) \cdot [\Delta(T_l - T_a) + P_{ws} - P_w] \tag{A10}$$

the sensible heat is computed as

$$H_l = c_H(a_{sH}, A_{tr}) \cdot [(T_l - T_a)] \tag{A11}$$

$A_{tr}$ is the transpiring surface for unit of ground surface [-], $a_{sE}$ are the sides of surface exchanging latent heat, equal to 1 for

hypostomatous, 2 for amphistomatous [-]; $a_{sH}$ are the sides of surface exchanging sensible heat and longwave radiation, equal to 1 for soil, 2 for leaves [-]; $P_{ws}$ and $P_w$ are the saturation water vapour pressure and the water vapour pressure. Eventually, the leaf temperature (for each layer treated) $T_l$ is computed as,

$$T_l = \frac{Rs + a_{sH} A_{tr} \epsilon_l \sigma T_a^4 + c_H(a_{sH}, A_{tr}) \cdot T_a + c_E(a_{sE}, A_{tr}, g_s) \cdot (\Delta e \cdot T_a + P_w - P_{ws})}{c_H(a_{sH}, A_{tr}) + c_E(a_{sE}, A_{tr}, g_s)\Delta e + a_{sH} A_{tr} \epsilon_l \sigma T^3} \tag{A12}$$

where $g_s$ is the stomatal conductance [$\text{m s}^{-1}$].

**A3   GEOET Classes in detail**

The refactoring of the GEOET codes cover some substantial software engineering aspects that are detailed below for some of the packages.

**Working details of the classe `geoet.data` and `geoet.inout`**

`Parameters`, `ProblemQuantities`, `InputTimeSeries` are the pivotal data classes that are used by algorithmic

classes to get the simulation done. Their relations to form a working program are illustrated in the UML diagrams presented in





**Figure A1.** UML class diagram for the `ProblemQuantities` class, illustrating the aggregation and association relations with the ET models solvers classes and mainly with the input and output classes. In this way we ensure that all model variables are shared among all classes regardless of the type of ET model used.






**Figure A2.** UML class diagram for the `InputTimeSeries` class, revealing the aggregation and association relations with some of the classes involved.





Figures 5, A1, A2. An identical pattern emerges in the relationships between the method ET solvers and the classes encapsulated within the `geoet.data` and `geoet.inout` packages. To illustrate this point, we will examine the connection between a solver class, such as *PriestleyTaylorActualSolverMain* and the *Parameters* class (Figure 5). The relationship between these classes can be considered both an *association* and an *aggregation*. It is an association because PriestleyTaylorActualSolver-

Main object uses the Parameters object only temporarily and is not responsible for its creation or lifecycle management. In fact, PriestleyTaylorActualSolverMain has a reference to Parameters to access its data, and Parameters exists independently of PriestleyTaylorActualSolverMain. But the relation is also an aggregation due to the class PriestleyTaylorActualSolverMain contains a field parameters of type Parameters, and this relationship indicates that PriestleyTaylorActualSolverMain "contains" an object of type Parameters. The important point is that the Parameters class is used by PriestleyTaylorActualSolverMain but

exists independently of it.



*Author contributions.* CD, RR and NT conceptualized the model's structure. NT implemented WHETGEO and several foundational classes crucial for the development of GEOSPACE-1D. Additionally, CD led the refactoring of ETP-GEOFRAME into GEOET, designed and implemented BrokerGEO and run the simulations. RR supervised the work of NT and CD and provided financial support. CD and RR collaborated on writing the paper and discussed any of its part.

*Competing interests.* There are no competing interest

*Disclaimer.* The software is provided under GPL 3.0 without any responsibility about its use.

*Acknowledgements.* The Authors were supported by the projects WATZON, and WATERSTEM. The project WATSON partially supported the travel and the stay of the first Author at EPFL.



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
