# Peer review of "A component based modular treatment of the soil-plant-atmosphere continuum: the GEOSPACE framework (v.1.2.9)"

_EGUsphere, 2024_

## Author Comment (AC1)

Dear Anonymous Referee #1,

Thank you very much for your review and your constructive comments. The entire text of your comment is shown (C) together with our authors' responses (A).

Kind regards,

Concetta D'Amato, Niccolò Tubini and Riccardo Rigon

**Reviewer #1**

**Title: A component based modular treatment of the soil-plant-atmosphere continuum: the GEOSPACE framework (v.1.2.9)**

**C1 - Summary:**

The manuscript introduces a novel modeling framework that focuses on modeling the interactions within the soil-plant-atmosphere continuum (SPAC). Rather than using a single, rigid model, the authors propose a modular approach called GEOSPACE-1D, built on object-oriented programming principles. GEOSPACE-1D is a flexible, open-source framework with self-contained components. This modular design allows for easy customization, reuse, and extension of the model without disrupting existing parts to integrate new processes seamlessly. Instead of offering a single definitive model. The manuscript thoroughly describes the framework's components, providing information about modularity, process representation and interaction between components. The authors further discuss the implementation of the system with a case study, describe the setup, and present the results.

A: Thank you for your thorough and accurate summary of the manuscript. We appreciate that you have highlighted the modular structure and object-oriented design of GEOSPACE-1D, as well as its potential for customization and integration of new processes. This feedback confirms that the intended framework structure and its key features have been clearly explained.

**General Comments:**

**C2 -** While supplementary materials provide detail on the GEOFRAME system, the main body could include a brief description of the system. Clearly establish GEOFRAME as the overarching framework and explicitly define GEOSPACE as the specific ecohydrological model within it. This clarification will significantly improve the reader's understanding of the present work and contextualize it.

A: We appreciate the reviewer's suggestion since it is important to clearly establish that GEOframe is a modular framework that allows users to build custom modeling solutions to

address various hydrological challenges, inside which GEOSPACE was born. GEOSPACE mainly represents the ecohydrological model of GEOframe due to it simulates the water transport in the soil-plant-atmosphere continuum, by using models available in GEOframe. We modified the abstract and introduction. Please refer to C9 and C10 comments.

**C3 -** Consider adding a brief introductory paragraph or section in the main body that explains the purpose and architecture of the GEOFRAME system, and how GEOSPACE fits within it.

A: We have modified the abstract and the introduction accordingly. Please see the C9 and C10 answers for more details.

**C4 -** The captions of the figures need to be more descriptive to meet the author's intention. Some figures have very descriptive captions and some others lack detail. Despite providing a description in the main body, it is important that the figures are self-explanatory or at least that the author helps the reader in their interpretation. Take, for example, Figure 3, where the width of the arrows is explained in the text but should also be included in the caption.

Instead of simply stating "[arrow] thickness reflecting the volume of exchanged variables" rewrite Figure 3's caption to include: "Arrow widths represent [specific meaning, e.g., water flow volume]."

A: The diagram depicts the component relationships within our modeling framework, highlighting parameter dependencies through weighted arrow representations. For instance, the "Stress Factor" component establishes connections with both Prospero components and ETBrokerSolver, delivering multiple parameters to these components (shown by arrows three times thicker than baseline). Conversely, it provides only a single parameter to the soil ET component, represented by a correspondingly thinner arrow. During model execution, WHETGEO initializes the computational sequence. The figure necessarily omits several auxiliary components that manage input/output operations and buffer processes essential for parallelizing component output writing, as including these would compromise diagram clarity and readability. Complete operational details are available in the simulation files (.sim files) included in the supplemental material. We direct reviewers' attention particularly to the `geospace1D_ProsperoPM.sim` file, which serves as an illustrative example of our framework's structure and functionality. Additionally, we have prepared a concise presentation with accompanying slides that demonstrate how .sim files are organized and interpreted which will be added to the supplementary material.

In response to reviewer feedback, we have substantially revised the manuscript text to better convey this information and expanded figure captions throughout the document to provide more comprehensive explanations of the modeling framework components and their interactions.

The text has been modified to include the following: "For a comprehensive understanding of the complete workflow, we direct readers to examine the simulation configuration files provided in the supplemental material, particularly the `geospace1D_ProsperoPM.sim` file. These .sim files, a standard feature of the OMS framework, serve as executable documentation that precisely records the model workflow, component connections, and parameter settings. The supplemental material also includes a concise presentation with accompanying slides that provide detailed guidance on interpreting .sim file structure, component relationships, and execution sequence, offering valuable insights for both new users and those seeking to modify existing simulations."

**C5** - While the number of figures in Section 7 is appropriate, the analysis and interpretation of the simulation results are insufficient. Provide a more thorough evaluation of the model's performance, including quantitative metrics (if possible) and qualitative assessments relative to the experiment setup.

A: The simulations presented here are not intended to investigate a particular experimental setup, which would require a longer paper and open a topic better addressed in dedicated research. Instead, these simulations serve as a proof of concept, demonstrating that the system functions correctly without integration errors across all tested conditions. This represents only a small subset of the comprehensive simulations performed by the Authors to evaluate GEOSPACE's robustness. Water budget closure was verified for all simulations, with supporting documentation provided in the supplemental material, even for challenging situations involving transitions between unsaturated and saturated conditions as apparent, for instance, in Figure 13. We have modified Section 7 to clarify these aspects and have furthermore improved all figure captions.

The text has been modified under the label C5, to include the following:"

To demonstrate the capabilities of the model, we present two "virtual"simulations: the first called as "baseline simulation" (BSL) which simulates the coupled dynamics of infiltration and evapotranspiration, while the second one focuses only on the infiltration process. The simulations are not intended to investigate a particular experimental setup, which would require a longer paper and open a topic better addressed in dedicated research. Instead, these simulations serve as a proof of concept, demonstrating that the system functions correctly without integration errors across all tested conditions. This represents only a small subset of the comprehensive simulations performed by the Authors  to evaluate GEOSPACE's robustness. Water budget closure was verified for all simulations, with supporting documentation provided in the supplemental material. All the simulations are readily available in the supplemental material which contain a series of Jupyter notebooks which allow to re-execute them and inspect their results more deeply. "

**C6** -Also, the authors could describe the processes represented in the simulations and explain the logic behind the observed model behavior. What can be considered limitations?

A:  Answering to this question is part of the modifications we made in section 7. Please, in the new manuscript with highlighted additions, refer to modifications with keyword C5.

**C7** - Consider improving the structure of the manuscript regarding sections and subsections. The paragraph preceding each subsection should provide a clear introduction and establish the connection to the subsequent content. For example, in Section 4, the unnumbered subsections (Priestley-Taylor ET estimator, Penman-Monteith FAO estimator, Prospero Model) should be introduced with a unifying paragraph that explains their relevance.

A: We modified the introduction to Section 4 to accomplish the Reviewer's requests. Please refer to C4 modifications.

**C8** - The transition into sections 4.1 and 4.1.1 should be handled with more clarity.

A: We have modified the initial part of section 4.1.1 by adding: "A primary objective in the software engineering of the GEOSPACE system was to enable feature expansion through class addition rather than code modification, adhering to the open-closed principle of object-oriented design."

**Specific Comments:**

**C9 - Abstract:** Include a couple of sentences describing the GEOFRAME system and the gap that GEOSPACE is filling. Add a mention of the processes that can be represented in the GEOSPACE framework.

A: We have modified the abstract by adding: "GEOSPACE leverages and extends selected components from the GEOframe modeling system, while also integrating newly developed modules to comprehensively simulate water transport dynamics in the SPAC system."

**C10 -** Introduction**:** Add details about the GEOFRAME system, and the relevance of including SPAC process representation. Include details on the performance of the framework when tested as presented in section 7. Clarify that GEOSPACE is a framework within the GEOframe system.

A: As mentioned previously, it is important to clearly establish that GEOSPACE is an integral component of GEOframe. The main text has been modified in the introduction to also specify the main contributions of this paper, as follow:

"The GEOSPACE framework functions as an integral component of the GEOframe system and it uses some of the components available in GEOframe to simulate the water transport

in the continuum SPAC, thus being the ecohydrological model of GEOframe. GEOframe is an open-source, component-based hydrological modelling system (Formetta et al., 2014; Bancheri et al., 2020). Rather than being a single model, GEOframe is a modular framework, where each part of the hydrological cycle is implemented in a self-contained building block, an OMS3 component (David et al., 2013). Models available in GEOframe cover a wide range of processes, including geomorphic and DEM analysis, spatial interpolation of meteorological variables, radiation budget estimation, infiltration, evapotranspiration, runoff generation, channel routing, travel time analysis, and model calibration. It allows users to build custom modeling solutions to address various hydrological challenges. The GEOSPACE framework presented here was developed by composing and extending existing GEOframe components: WHETGEO (Water Heat and Transport) (Tubini and Rigon, 2022), GEOET (EvapoTranspiration), and BrokerGEO to simulate complex soil-vegetation-atmosphere interactions in the Critical Zone. While GEOSPACE builds upon existing components in GEOframe, this work contributes three main innovations: (i) the development of GEOET, a new evapotranspiration module evolved from the established ETP-GEOframe component (Bottazzi, 2020); (ii) the implementation of BrokerGEO, a new coupler component enabling the dynamic interaction between evapotranspiration and infiltration processes; (iii) the extension of WHETGEO (Tubini and Rigon, 2022) to allow modular and seamless coupling with GEOET and BrokerGEO. These contributions represent both algorithmic and structural advances over previous models, such as the monolithic GEOtop framework (Rigon et al., 2006), and establish GEOSPACE as the ecohydrological core of GEOframe."

In addition, a reference to Section 7 simulations was included in the introduction as requested by the reviewer. Modifications were made in the introduction under Keyword C9

**C11 - GEOSPACE-1D System Overview and its perceptual model -** There is a mention to "*multiple stress functions mentioned in the introduction*" but such reference is missing in the introduction

A: OK. We have corrected it by deleting the sentence.

**C12 General notes about the software organization of GEOSPACE-1D -** Figure 3 could be improved by including a description/functionality of each component within the SPAC. The thickness of the arrows represents the number of variables, but it is not described in the caption or the number. Is there a sequence in the computing of each component, if so, is there a starting point?

A:  The modification of the caption was accomplished in the revised  manuscript. Please see the C4 answer.

**C13 - GEOET**

Considering improving the structure of this section. **The Priestley-taylor ET estimator; The Penman-Montheith FAO estimator; The Prospero Model**

A: According to what already said in answering C7 and C8 we have modified the introduction of the section and the text in the revised manuscript.

**C14 - The GEOET informatics organization- How to add a new model?:** Is this section only referring to new models for GEOET?

A: Thank you for the question. The section is not limited to models for GEOET, but refers to the integration of new models into any component of the GEOSPACE framework. We have clarified this point in the revised manuscript under label C8.

**C15 - Unveiling GEOSPACE-1D capabilities on practical applications -** Figure 12 is not very clear. Consider using a different scale or color scheme to better show the temporal variability. Additionally, incorporating the rainfall timeseries could help interpret the variation shown in this figure while tracking the the occurrence of rainfall events. Consider providing a side by side comparison between the most relevant aspects of the two experiments.

We thank the Reviewer for this valuable comment. Following the suggestion, we have substantially revised Figure 12 to improve clarity and interpretability. Specifically:

1. We incorporated the precipitation–irrigation time series as Panel (a), which now allows direct temporal comparison between rainfall inputs and soil water potential responses in the two simulations.
2. We ensured that the color scales are consistent and diverging where appropriate: Panel (b) presents absolute soil water potential ($\psi$) with a perceptually uniform scale, while Panel (c) uses a diverging colormap centered at zero to effectively highlight both positive and negative differences in $\Delta\psi = \psi_R - \psi_{BSL}$.
3. To enhance the comparison between scenarios, we now show:
   - The full spatio-temporal evolution of $\psi$ in the baseline case (Panel b),
   - The direct difference between the two scenarios (Panel c), thus enabling a side-by-side assessment of the processes driving divergence (e.g., absence of evapotranspiration in the infiltration-only simulation).

We also revised the figure caption to better explain the physical meaning of the color scales and the relevance of the differences. Hopefully this meets the Reviewer's request.

**C16 - User information: Input and output -** This information should be included in code and data availability. Section 8

A: We appreciate the reviewer's suggestion. To improve clarity regarding the open-source structure and usage of the code, we decided to separate the information related to input and output into a dedicated section (Section 8) rather than incorporating it into the Code and Data Availability section. This choice was made to provide users and developers with a more accessible and detailed overview of how the components interact at runtime, including input requirements, output formats, and integration aspects. We believe this structural decision enhances readability and usability, while preserving the coherence of the manuscript.

---

## Author Comment (AC2)

Dear Anonymous Referee #2,

Thank you very much for your review and your constructive comments. The entire text of your comment is shown (C) together with our authors' responses (A).

Kind regards,

Concetta D'Amato, Niccolò Tubini and Riccardo Rigon.

**Reviewer #2**

**Title: A component based modular treatment of the soil-plant-atmosphere continuum: the GEOSPACE framework (v.1.2.9)**

**Summary**

**C17** - This paper presents a modeling framework with three sub-components aiming to improve the simulation capabilities of soil-plant-atmosphere continuum. The paper primarily focuses on presenting the software rather than the science or the specific results.

A: We believe that presenting the software aligns perfectly with the journal's scope. Although it may appear reductive, we contend that meaningful progress in hydrological modeling requires software developed with proper engineering principles. Poor or inaccurate implementations can obscure the underlying physics they aim to represent. Additionally, sound software design enhances code readability, maintainability, and facilitates thorough inspection—a crucial aspect of modern scientific practice. Expanding the paper to include real case studies would have broadened its scope excessively and increased its already substantial length. The current paper is quite comprehensive, and adding approximately 20 pages of case studies, in our opinion, would likely discourage readership. Modifications were made in the introduction under Keyword C17 to convey these concepts.

**C18** - I've checked the overall paper for clarity and some of the formulations, but I haven't checked the math in depth since that would be a little out of my domain. I'd recommend that it gets checked in other parts of the review. However, it seems sub-models have been already published and this paper is more focused on the integration.

A:Several GEOSPACE components have undergone substantial modifications beyond their previously published versions. As illustrated in Section 4, GEOET's refactoring represents necessary architectural changes enabling overall integration and future code expansion, not merely cosmetic improvements. Similarly, portions of WHETGEO were modified to serve the same objectives. The treatment of root functioning and evolution, while elementary, is entirely new, as is the BrokerGEO software that facilitates feedback among SPAC components. These contributions have been emphasized in the Conclusions section.

**C19** - One overall major comment that didn't fit in the section is that it would help to produce a table of similar class of models, their short descriptions, and key features evaluated against key advances in GEOSPACE-1D.

A: We appreciate the reviewer's suggestion for comparative tables. While understandable, comprehensive model comparisons are more suited to dedicated review articles. For such comparisons, we refer readers to Blyth et al. (2021), Fisher and Koven (2020), and Pal and Sharma (2021) where the Reviewer can also find the Tables they search for. Nevertheless, we have added relevant references to provide additional context in the Introduction.

Model selection involves multifaceted considerations beyond scientific capabilities. These include software architecture, licensing, extensibility, and implementation languages—factors critical to our development decisions. For instance, while HYDRUS is well-validated and widely used in agro-climatology, its FORTRAN implementation and commercial licensing for 2D/3D versions constrained our adoption. Additionally, HYDRUS-1D has technical limitations regarding ponding formation, Richards equation integration, and vegetation representation that we aimed to improve. Similarly, the Community Land Model, though developed by leading researchers, employs software architecture that limits the flexible, component-based modeling framework we envision. Other established models like JULES, ORCHIDEE, and NOAH, despite their valuable features, have grown increasingly complex. This complexity makes understanding, testing, and modifying their implementations difficult—challenges we experienced firsthand with GEOtop, which became unmanageably complex over time. These experiences motivated our shift toward a contemporary, component-based modeling infrastructure that enables better separation of concerns and software accountability. This approach maintains complete control over code evolution while creating a unified framework applicable to both agro-meteorological and hydro-climatological communities. For readers interested in comprehensive model overviews, we recommend Blyth et al. (2021) for LSM comparisons, Fatichi et al. (2016) for process-based model capabilities, and Bonan et al. (2024) for Earth System modeling perspectives. Additional valuable references include Overgaard et al. (2006), McDermid et al. (2017), Vereecken et al. (2019), Bierkens et al. (2015), Graeme et al. (2023), and Miralles et al. (2024).

We have expanded the Introduction to incorporate these considerations.

References

Bierkens, Marc F. P. 2015. "Global Hydrology 2015: State, Trends, and Directions." *Water Resources Research* 51 (7): 4923–47. https://doi.org/10.1002/2015WR017173;

Bonan, Gordon B., Oliver Lucier, Deborah R. Coen, Adrianna C. Foster, Jacquelyn K. Shuman, Marysa M. Laguë, Abigail L. S. Swann, et al. 2024. "Reimagining Earth in the Earth System." *Journal of Advances in Modeling Earth Systems* 16 (8). https://doi.org/10.1029/2023ms004017.

Endrizzi, S., S. Gruber, M. Dall'Amico, and R. Rigon. 2014. "GEOtop 2.0: Simulating the Combined Energy and Water Balance at and below the Land Surface Accounting for Soil Freezing, Snow Cover and Terrain Effects." *Geoscientific Model Development* 7 (6): 2831–57. https://doi.org/10.5194/gmd-7-2831-2014.

Stephens, Graeme, Jan Polcher, Xubin Zeng, Peter van Oevelen, Germán Poveda, Michael Bosilovich, Myoung-Hwan Ahn, et al. 2023. "The First 30 Years of GEWEX." *Bulletin of the American Meteorological Society* 104 (1): E126–57. https://doi.org/10.1175/bams-d-22-0061.1.

Miralles, Diego G., Jordi Vilà-Guerau de Arellano, Tim R. McVicar, and Miguel D. Mahecha. 2025. "Vegetation-Climate Feedbacks across Scales." *Annals of the New York Academy of Sciences* 1544 (1): 27–41. https://doi.org/10.1111/nyas.15286.

Overgard et al 2006; McDermid, S. S., L. O. Mearns, and A. C. Ruane. 2017. "Representing Agriculture in Earth System Models: Approaches and Priorities for Development." *Journal of Advances in Modeling Earth Systems* 9 (5): 2230–65. https://doi.org/10.1002/2016ms000749;

Rigon, R., G. Bertoldi, and T. Over. 2006. "GEOtop: A Distributed Hydrological Model with Coupled Water and Energy Budgets." *Journal of Hydrometeorology* 7 (June): 371–88. https://doi.org/10.1175/JHM497.1.

Vereecken, Harry, Lutz Weihermüller, Shmuel Assouline, Jirka Šimůnek, Anne Verhoef, Michael Herbst, Nicole Archer, et al. 2019. "Infiltration from the Pedon to Global Grid Scales: An Overview and Outlook for Land Surface Modeling." *Vadose Zone Journal: VZJ* 18 (1): 1–53. https://doi.org/10.2136/vzj2018.10.0191;

**C20** - Other major comments include no assets being available for review/links not working, big wiring diagram is missing, case implementation and validation could be improved, performance metrics to be included etc. Our detailed comments, including minor and major, are organized below in the order of appearance in the manuscript. Most of these comments are going to be applicable throughout the manuscript but I've highlighted them at only a few places.

A: We appreciate the reviewer's detailed comments and address each point below:

• Review/links not working: We have systematically reviewed all manuscript links and verified their functionality in the revised version.

• Missing wiring diagram: Figure 3, whilst a simplification, is such a diagram. Please see answer to #1 reviewer C4 for more details.

• Case implementation and validation: As previously discussed (Response C5 to the first Reviewer), implementing and validating a detailed case study represents a substantial undertaking that could merit a separate publication. Including a comprehensive case study would add approximately 20 pages to an already extensive manuscript (50 pages, expanding to hundreds with supplementary material). We would be happy to address any specific concerns about our current case implementation.

• Performance metrics: We maintain that mass conservation during integration is the most relevant performance metric for this paper, as previously detailed. Code efficiency comparisons would require benchmarking against other software packages, which—as noted in Response C19—would be both technically challenging and potentially inappropriate given the differences in software architectures and objectives.

We are committed to addressing the reviewer's concerns thoroughly and look forward to more specific feedback in the detailed comments below.

**Specific comments**

Abstract

**C21** - Line 3 - specify matter? do you mean organic matter?

A: Thank you for pointing this out. We have removed the word "matter" from the sentence to avoid ambiguity.

**C22** - Line 4 and throughout the text: deemphasis interdisciplinary aspects. This paper is still a very specific product of ecohydrologists, without any input from, let's say, economists. It is fine to mention the need for interdisciplinary science and products but probably don't frame this as an interdisciplinary product.

**A**: Thank you for the comment, as it likely indicates that our perspective was not clearly conveyed. What we aim to emphasize is that modeling the soil-plant-atmosphere system encompasses a wide range of disciplines, including hydrology, ecology, meteorology, climatology, geology, agronomy, environmental chemistry, environmental engineering, remote sensing, and, not least, numerical modeling and computational science. Given this context, a physically-based approach to studying the SPAC system inherently requires an interdisciplinary intellectual effort. To better clarify this concept, the text has been revised as

follows: "Modeling the SPAC system involves multiple disciplines, including hydrology, ecology, and computational science, making a physically-based approach inherently interdisciplinary and essential for capturing the complexity of the system."

**C23 -** Line 24 – It is important to make sure it produces some kind of results. Or a prototype model or case implementation is critical for this paper to be strong. I see Section 7 speaks to it but there are some concerns there as described later. Also, better to mention the case implementation and discuss some results/over model behavior here in the abstract as well.

A: We thank the reviewer for highlighting this important point. In response, we have revised the abstract to include a clear reference to the case implementation. Specifically, we now mention the virtual simulations presented in Section 7, which demonstrate the model's ability to simulate the coupled dynamics of infiltration and evapotranspiration within the soil–plant–atmosphere continuum. These additions aim to strengthen the overall contribution of the work and emphasize the operational capabilities of the GEOSPACE-1D framework. Moreover, to understand better we'll look at the concerns that follow.

**C24 -** Intro

It is unclear from the introduction what this paper is contributing. I see many motivations being described such as better SPAC modeling, going beyond traditional "models", MBC etc, but the description of the unique contribution is lacking. I'd suggest not only specifying that comprehensively but also including an overarching "vision" statement for the model covering its scope, specifications and significance.

A:  This approach (i.e., MBC) maintains complete control over code evolution while creating a unified framework applicable to eco-hydrological, agro-meteorological, and hydro-climatological communities. In addressing these concepts, we also aim to develop codes that strictly adhere to FAIR principles of openness and availability. Last, but not least, we also address algorithmic limitations that we have identified in similar software, such as inadequate treatment of transitions between saturated and unsaturated conditions, and the use of improper solvers. We have added to the new manuscript text to convey these concepts.

Section 2

**C25 -** Line 108 and throughout the text – "with flexibility and minimal effort": Major: Since this is primarily a software paper, I'd like to see performance metrics included in the SI or the appendices. A comparison with other models/software or previous versions would also be nice. It is understandable if those are not available for other models but a comparison to the preceding version would help the reader see the value of this contribution more clearly.

A:  Benchmarking against other software is in our opinion a matter for specialized community papers that focus on standardized benchmark simulations. Since this software is novel in parts of  its components and their connections, there is no previous version to compare with regarding execution velocity. The benchmarks we have performed focus on water mass budget conservation. The value of this contribution is described more clearly in our responses to comments C19.

**C26 -** Line 114 and throughout the text – I see a few critical citation are referencing to authors own previous work. No issues with that but it would make the paper stronger if some of the formulations/key statements could also be supported by other citations. Just a suggestion

A:  The implementation of the Casulli-Zanolli algorithm is not only unique to WHETGEO (and GEOSPACE) thus far, but according to the original authors, apparently represents the only method that guarantees solver convergence in all cases without requiring external controls, like, for instance in our GEOtop model (e.g Endrizzi et al, 2014) that uses a Newton-Krylov method. At line 114 we have added a reference to the Casulli and Zanolli paper (Casulli and Zanolli, 2010).  Because we are convinced that the statement made by the mathematicians we cite, while strong, is true, we are reluctant in this specific case to cite other papers, such as Celia 1990 or Paniconi and Putti 1994, that apparently use incomplete integration methods. Moreover, models like CATHY (Paniconi and Putti), while certainly valuable, depend on ad hoc treatments of surface saturated and ponding conditions that we avoid. However, we have added additional citations to other authors where appropriate.

References

Casulli, Vincenzo, and ZANOLLI. 2010. "A Nested Newton-Type Algorithm for Finite Colume Methods Solving Richards' Equation in Mixed Form." *SIAM Journal of Scientific Computing* 32 (4): 2225–73.

Celia, M. A., E. T. Bouloutas, and R. L. Zarba. 1990. "A General Mass-Conservative Numerical Solution for the Unsatured Flow Equation." *Water Resources Research* 26 (7): 1483–96.

Endrizzi, S., S. Gruber, M. Dall'Amico, and R. Rigon. 2014. "GEOtop 2.0: Simulating the Combined Energy and Water Balance at and below the Land Surface Accounting for Soil Freezing, Snow Cover and Terrain Effects." *Geoscientific Model Development* 7 (6): 2831–57. https://doi.org/10.5194/gmd-7-2831-2014.

Paniconi, C., and M. Putti. 1994. "A Comparison of Picard and Newton Iteration in the Numerical Solution of Multidimensional Variably Saturated Flow Problems." *Water Resources Research* 30 (12): 3357–3333. https://doi.org/10.1029/94WR02046.

**C27** - Fig 1 - Are these components developed as a part of this effort/paper. it is not clear so far

A:  We have clearly specified in the revised manuscript which contributions are developed in this paper, beginning with the introduction. The primary innovation presented is the establishment of bidirectional connectivity between infiltration and transpiration processes, enabling dynamic feedback mechanisms between soil conditions and atmospheric states. To achieve this integration, significant portions of the existing software were refactored, particularly the transpiration modules, and to a lesser extent, the infiltration components. The root growth and behavior modeling framework represents an entirely novel contribution. We have added the following clarification to the new manuscript introduction:

*"The GEOSPACE framework presented here was developed by composing and extending existing GEOframe components: WHETGEO (Water Heat and Transport) (Tubini and Rigon, 2022), GEOET (EvapoTranspiration), and BrokerGEO to simulate complex soil-vegetation-atmosphere interactions in the Critical Zone. While GEOSPACE builds upon existing components in GEOframe, this work contributes three main innovations: (i) the development of GEOET, a new evapotranspiration module evolved from the established ETP-GEOframe component (Bottazzi, 2020); (ii) the implementation of BrokerGEO, a new coupler component enabling the dynamic interaction between evapotranspiration and infiltration processes; (iii) the extension of WHETGEO (Tubini and Rigon, 2022) to allow modular and seamless coupling with GEOET and BrokerGEO. These contributions represent both algorithmic and structural advances over previous models, such as the monolithic GEOtop framework (Rigon et al., 2006), and establish GEOSPACE as the ecohydrological core of GEOframe."*

**C28 -** Line 136 – Major comment: There are many modules/components within the GEOframe suite. I see Fig 1 and Fig 3 attempt to list a few but a bigger wire-diagram showing all components with their connection is critically needed to follow what's going on and how everything works together. I suggest including that as a separate Fig at the start. It is fine if that fig gets complex, sometimes looking everything in one place is much better than trying to connect across pages

A: Same as in C20: While we could create a comprehensive wiring diagram, it would be extremely large and potentially counterproductive to readability. Instead, we direct interested readers to the geospace1D_ProsperoPM.sim file, which provides a readable workflow representation of the entire model. This file is included in the supplementary material, with clear comments to help readers understand the implementation. Besides,  we

have prepared a concise presentation with accompanying slides that demonstrate how .sim files are organized and interpreted.

**C29 -** Section 4

Major: In this section, I would suggest highlighting the strengths of the formulation in this paper compared to Penman-Monteith and Priestley-Taylor since you describe them as simplified approaches in the intro. "Traditional PBM-based land surface models, widely used in hydrology and agronomy, often employ simplified governing equations, such as the Penman-Monteith equation (Pereira et al., 2015) or the Priestley-Taylor approach". Alternatively, you can repurpose to better highlight Prospero.

A:  As suggested by the Reviewer, we have repurposed the Prospero model and more clearly specified its implementation details at the beginning of Section 4. We have included reference to a recent paper co-authored by two members of our team (D'Amato and Rigon, 2025)  that provides a comprehensive derivation of Prospero's governing equations and thoroughly addresses its limitations.

Reference

D'Amato, Concetta, and Riccardo Rigon. 2025. "Elementary Mathematics Helps to Shed Light on the Transpiration Budget under Water Stress." *Ecohydrology: Ecosystems, Land and Water Process Interactions, Ecohydrogeomorphology* 18 (2). https://doi.org/10.1002/eco.70009.

**C30** - Line 334: "GEOET, developed as part of this paper": Too far into the paper to mention this. Suggest being upfront about the key unique contributions of the model/paper

A: We appreciate the reviewer's suggestion. In the revised manuscript, we have anticipated the description of the key contributions of this work by including them in the Introduction section, as recommended. Specifically, we now state:

*"While GEOSPACE builds upon existing components in GEOframe, this work contributes three main innovations: (i) the development of GEOET, a new evapotranspiration module evolved from the established ETP-GEOframe component  (Bottazzi, 2020; (ii) the implementation of BrokerGEO, a new coupler component enabling the dynamic interaction between evapotranspiration and infiltration processes; (iii) the extension of WHETGEO (Tubini and Rigon, 2022) to allow modular and seamless coupling with GEOET and BrokerGEO. These contributions represent both algorithmic and structural advances over previous models, such as the monolithic GEOtop framework  (Rigon et al., 2006), and establish GEOSPACE as the ecohydrological core of GEOframe."*

We believe this change clarifies the scope and novelty of the paper from the outset, in line with the reviewer's recommendation.

**C31 -**Lines 560-575: non-critical writing style check: suggest making proper paras

A: In the revised text we have modified the structure of the paragraphs. Hopefully now, everything is more readable.

**C32 -**Page 30 and Figs 12-19 – Major: no validation of any sorts is presented to confirm the behavior of the model's outputs. I'd highly recommend a comparison with data and in the worst case an expert-based evaluation of the model diagnostics.

A:  We are addressing applications of our model in separate forthcoming papers, as the experimental setups require detailed descriptions, particularly regarding evapotranspiration measurements, which often present their own methodological challenges and uncertainties. However, following Reviewer #1's suggestion in comment C5, we have incorporated additional expert commentary throughout the manuscript to enhance the interpretability of our simulation results and make their significance more accessible to readers.

**C33 -** No assets were available for review. SPIKE II data is available on zenodo but no links in Section 8 or the code available are functions. I would recommend including them as texts too: https://github.com/geoframecomponents/GEOSPACE-1D

A: We thank the reviewer for this valuable observation. In the revised manuscript, we have provided all relevant links in full, both in Section 8 and in the Code and Data Availability section. These include direct access to the GEOSPACE-1D code repository, the corresponding OMS3 project, the Zenodo archive containing all materials required to reproduce the simulations and the SPIKE II experimental dataset. All assets are now explicitly referenced and accessible to ensure full transparency and support reproducibility.

---

## Author Comment (AC3)

(a) Precipitation–irrigation input over time.

[Figure]

(b) Soil water potential behavior in the baseline simulation.

[Figure]

(c) Temporal evolution of the soil water potential difference along the soil profile, $\Delta\psi = \psi_R - \psi_{BSL}$, where $\psi_R$ refers to the simulation with infiltration only, and $\psi_{BSL}$ to the baseline simulationa.

**Figure 12.** C15 Comparison of soil water potential evolution under scenarios with and without evapotranspiration: Panel (a) shows the precipitation–irrigation input over time. Panel (b) depicts the soil water potential ($\psi$) in the baseline simulation, which includes both infiltration and evapotranspiration. The plot displays depth in meters, with a color scale representing $\psi$. C5-C32 The darker blue colors indicate increases in $\psi$ resulting from rainfall events and their subsequent propagation over time, typically in a downward direction. As depth increases and water potential decreases, infiltration rates slow down due to reduced soil hydraulic conductivity. This phenomenon is visually represented by the decreasing intensity of the darker signatures and their rightward curvature with depth. Panel (c) illustrates the temporal evolution of the soil water potential difference, defined as $\Delta\psi = \psi_R - \psi_{BSL}$, where $\psi_R$ refers to the simulation with infiltration only, and $\psi_{BSL}$ to the baseline simulation. A diverging colormap centered at zero is used to represent $\Delta\psi$ values. Positive differences (blue) indicate higher $\psi_R$ in the infiltration-only case, while negative differences (red) indicate lower $\psi_R$. These deviations primarily result from the absence of evapotranspiration in the infiltration-only simulation.